# Comparative Effect of Frying and Baking on Chemical, Physical, and Microbiological Characteristics of Frozen Fish Nuggets

**DOI:** 10.3390/foods10123158

**Published:** 2021-12-20

**Authors:** David Oppong, Worawan Panpipat, Ling-Zhi Cheong, Manat Chaijan

**Affiliations:** 1Food Technology and Innovation Research Center of Excellence, School of Agricultural Technology and Food Industry, Walailak University, Nakhon Si Thammarat 80160, Thailand; oppondave@gmail.com (D.O.); pworawan@wu.ac.th (W.P.); 2Zhejiang-Malaysia Joint Research Laboratory for Agricultural Product Processing and Nutrition, College of Food and Pharmaceutical Science, Ningbo University, Ningbo 315211, China; cheonglingzhi@nbu.edu.cn

**Keywords:** cassava croaker, quality, storage, healthier food, lipid oxidation

## Abstract

The effects of deep-frying and oven-baking on chemical, physical, and microbiological, properties of cassava croaker (*Pseudotolithus senegalensis*) nuggets during frozen storage were investigated. The moisture, protein, fat, ash, and carbohydrate contents varied among the cooking methods and frozen storage times. The deep-fried nugget had a higher fat content, which resulted in a higher energy value (*p* < 0.05). The free fatty acid content and peroxide value (PV) of the oven-baked nuggets were higher than the deep-fried ones (*p* < 0.05). The PV tended to increase with increasing storage time, but it was still within the recommended range for consumption. The deep-fried nugget showed a vivid orange–yellow color, with higher *L**, *a**, and *b** values, while oven-baked nuggets showed a pale-yellow color. The baked nuggets had relatively lower total expressible fluid than the deep-fried nuggets at all time points (*p* < 0.05). The hardness, springiness, and chewiness of deep-fried nuggets were higher than baked nuggets throughout the storage period (*p* < 0.05). The total plate count and yeast and mold counts produced by the two cooking methods were within the acceptable range throughout the storage.

## 1. Introduction

Globally, there has been a gradual expansion of interest in health and healthy eating, which has prompted the food industry to produce or adapt food preparations in this direction [1]. Consumption of functional components found in marine fish-based products, particularly *n*-3 polyunsaturated fatty acids such as eicosapentaenoic acid and docosahexaenoic acid, is rapidly increasing in order to improve consumer health and prevent disease including arteriosclerosis and thrombosis [2]. Additionally, fish is a rich source of proteins, amino acids, vitamins, and minerals [3,4]. Cassava croaker or cassava fish (*Pseudotolithus senegalensis*) is an economically important marine fish of tropical countries, particularly in West Africa [5]. It belongs to the high-protein, lean fish category (21.2% protein and 0.9% fat) and is a source of minerals (calcium, potassium, sodium, phosphorus, magnesium, iron, zinc, and manganese) [6]. Thus, this fish has good nutritional value and can benefit human nutrition.

Although the fish have high nutritional value, the time for their preparation and limited shelf life may discourage some consumers to purchase, leading to a preference for convenience products [7]. Fish nugget production is one of the fish processing techniques that can increase the value and reduce the postharvest loss in the fishing industry. It is regarded as an alternative to increasing the consumption of fish and fish-based food products because of global changes in consumer lifestyle. Fish nuggets are a modern food produced from fish fillets or ground fish meat, with spices, binders such as eggs, corn starch, various flours, and milk solids, and then coated with a coating [8]. Commercially, the fish nuggets, either raw or precooked, are generally in the frozen form because freezing and cold storage are effective methods of seafood preservation [9,10,11].

Fish-based food products can be cooked by several methods to retain and/or improve the quality of the final product, and nutritional value and cooking also positively associated [12]. Several physical alterations and chemical reactions occur during cooking, which can change the food nutritional component [13,14]. Deep frying and baking of fish and/or meat are regarded as two of the oldest food processing methods that impart quality characteristics to the food. In general, different types of cooking and processing can damage key nutrients in foods to varying degrees. As a result, it is an important to check how well different cooking methods preserve nutritional and other quality attributes in fish nuggets. There is no information regarding the quality aspects of deep-fried and oven-baked cassava fish nuggets during frozen storage. Therefore, the present study aimed to investigate the effects of deep-frying and oven-baking on the quality characteristics of frozen (−18 °C/90 days) cassava fish nuggets.

## 2. Materials and Methods

### 2.1. Ingredients and Chemicals

All of the ingredients for the fish nuggets such as garlic, ginger, seasoning, black pepper, parsley, mustard, lemon, and salt were purchased from a local supermarket in James Town, Accra, Ghana. All analytical chemicals were purchased from Sigma–Aldrich (St. Louis, MO, USA).

### 2.2. Fish Fillet Preparation

Ten cassava croaker (*P. senegalensis*) weighing 1–2 kg were purchased from a local market in James Town, Greater Accra, Ghana, West Africa. The fishes were placed in ice with a fish/ice proportion of 1:2 (*w*/*w*) and transported to the Laboratory of Food Research Institute, Accra, Ghana, within 50 min. Then, the fish were immediately washed with cold tap water (4 °C), gutted, clean, and filleted. The fish fillets were cut into 17–18 g square shapes (4 × 4 × 1.5 cm; width × length × height).

### 2.3. Nugget Preparation from Fish Fillet

The nuggets were produced using 3.2 kg of fish fillets for each treatment (deep-frying and oven-baking). For each treatment, approximately 182 of cut fillet cubes (CFC) were used. CFC were mixed with 70.6 g of garlic, 96.7 g of ginger, 27.8 g of ready-made seasoning powder (Knorr^®^, Bangkok, Thailand), 7.5 g of black pepper, 3.2 g of parsley, 37.5 g of mustard paste, 15.5 g of lemon juice, and 16.1 g of salt. The seasoned CFC were later dipped in 274.8 g of wheat flour. The floured CFC were dipped into 286.6 g of egg white, allowing any excess to drip off, and then rolled in 357.1 g of dried breadcrumbs. The nuggets were kept on the tray for 30 min to allow the coating to set. According to AOAC [15], the proximate composition of the raw nuggets was 55.3 ± 0.9% moisture, 34.0 ± 0.1% protein, 5.9 ± 0.1% fat, 2.5 ± 0.1% ash, and 2.3 ± 0.8% carbohydrate. Raw nuggets (unfrozen, Day 0) were either oven-baked or deep-fried, and the quality parameters were determined. For the frozen storage, raw nuggets were kept on plastic trays and subjected to blast freezing at −21 °C for 1 h using a Polar DN494 367 Blast Freezer (Campbelltown, NSW, Australia). The fish nuggets were then stored in Ziploc bags and frozen storage at −18 ± 2 °C for 90 days. A voltage stabilizer was supplied with the freezer to prevent temperature fluctuations in the case of power failure. Frozen samples were taken for deep-frying or oven-baking, after partially thawed for 60 min at 4 °C, at 15 days intervals, and further analyses were carried out. At each time point, 26 nuggets were randomly taken for analysis, and each test was done in triplicate.

### 2.4. Deep-Frying and Oven-Baking of Nuggets

One part of the fish nuggets (26 nuggets) was deep-fried in 5-L King Rice Bran Oil (γ-oryzanol, 8000 mg/kg) (Thai Edible Oil Co., Ltd., Bangkok, Thailand) at a temperature of 160 ± 5 °C [16] for 6 min in a 6-L Shuangchi electrical fryer pan (Guangzhou Shuangchi Dining Equipment Co., Ltd., Guangzhou, China) with a food-to-oil ratio of 1:10 (*w*/*v*). The second portion of fish nuggets (26 nuggets) was oven-baked at a temperature of 160 ± 5 °C for 45 min using a JY-OE60T5 electric convection oven (Zhongshan Jiaye Electric Appliance Co., Ltd., Guangdong, China) [17]. The deep-fried nuggets were placed in a frying basket for draining out the oil, whereas the oven-baked nuggets were placed on baking paper trays for further cooling and they were both analyzed for quality parameters. Due to the differences in heating medium, different cooking times were used for both types of nuggets in order to achieve the desired quality in the finished products.

### 2.5. Proximate Composition and Energy Value

The standard methods of AOAC [15] were used for proximate composition analysis including moisture, crude protein (a conversion factor = 6.25), fat, ash, and carbohydrate (calculated by difference). The energy value was determined using the Equation (1) adopted from Bonfim et al. [18]:(1)$$\mathrm{Energy}\;\mathrm{value}\;\left(\mathrm{kcal}/100\;g\right)=\left[4\times \mathrm{protein}\left(\%\right)\right]+\left[9\times \mathrm{fat}\left(\%\right)\right]+\left[4\times \mathrm{carbohydrate}\left(\%\right)\right]$$

### 2.6. Free Fatty Acid (FFA) and Peroxide Value (PV)

Lipid was extracted from processed fish nuggets by the method of Bligh and Dyer [19] and used for the determination of FFAs and PV. The FFA content was quantified using the Lowry and Tinsley method [20]. Briefly, isooctane (5 mL) and 5% cupric acetate–pyridine (1 mL) were added to a fat sample (0.1 g) and vortexed for 90 s. The absorbance of the top phase was measured at 715 nm using a UV-Vis spectrophotometer (Shimadzu, Kyoto, Japan). The FFA content was estimated using a standard curve of oleic acid [20]. For PV, AOCS Method Cd 8b-90 [21] was used and reported as milliequivalents of free iodine (meq)/kg fat.

### 2.7. Color and Total Expressible Fluid (TEF)

The *L***a***b** color parameters of processed fish nugget samples were examined with a Hunterlab Miniscan/EX instrument (Hunter Assoc. Laboratory Inc., Reston, VA, USA).

For the TEF, two cooked nuggets (25 g each) (Ws) were placed in a preweighed centrifuge tube and then centrifuged (2960× *g*/25 °C/3 min) using an RC-5B plus centrifuge (Sorvall Inc., Norwalk, CT, USA). The pelleted samples were removed and weighed (Wp). Then, the TEF was calculated using Equation (2) [22].
(2)$$\mathrm{TEF}\;\left(\%\right)=\frac{\mathrm{Ws}-\mathrm{Wp}}{\mathrm{Ws}}\times 100$$

### 2.8. Texture Profile Analysis (TPA)

TPA of nuggets was examined using a TA-XT plus texture analyzer (Stable Micro System, Godalming, UK) fixed to texture expert software following the methods described by Bonfim et al. [18]. Texture analysis was performed using nuggets sizes of 1.5 × 1.5 cm, and the compression was done twice to 80% of the original height using a compression probe (P 36) at a speed of 2.0 mm/s and a time of 1.00 s. The hardness, springiness, cohesiveness, and chewiness were recorded.

### 2.9. Microbiological Analysis

Total plate count (TPC) and yeast and mold counts of deep-fried and baked fish nuggets were enumerated as per Panpipat and Chaijan [23], over 90 days, at 15-day intervals.

### 2.10. Statistical Analysis

A completely randomized design was used, and the experiment was triplicated. Data were subjected to one-way analysis of variance. Mean comparison was performed by Duncan’s Multiple Range Test using the SPSS program (Version 23.0, SPSS Inc., Chicago, IL, USA). The *t*-test was used in the case of pairwise comparison.

## 3. Results and Discussion

### 3.1. Proximate Composition and Energy Value

The effects of deep-frying and oven-baking on the proximate compositions and energy values of cassava fish nuggets are shown in Table 1. The moisture, protein, fat, ash, and carbohydrate contents varied among the cooking methods and frozen storage times. In fresh nuggets (Day 0), oven-baked nuggets had a higher moisture content, while deep-fried nuggets had a higher fat content (*p* < 0.05). The results obtained were comparable to those published by Marimuthu et al. [24]. During deep-frying, water was evaporated and partially replaced by the diffusion of the frying oil into the fish nugget, leading to a higher fat content in deep-fried fish than in oven-baked fish [24,25,26,27]. No significant differences in protein, ash, and carbohydrate contents in nuggets at Day 0 were noticeable among the two cooking methods (*p* > 0.05). During storage, the moisture and carbohydrate content decreased with associated increase in protein, fat, and ash content in both cooking methods. Moisture loss, due to dehydration, during frozen storage may explain the decrease in moisture content of the cooked nuggets. At the same time, the loss of breadcrumbs from the coating during frozen storage and cooking may cause the decrease in carbohydrate content of the cooked nuggets. Loss of coating can occur at any stage during the preparation or frozen storage [28]. The protein content was higher in oven-baked nuggets than in deep-fried nuggets from Day 15 to Day 90 of storage. The fat content in deep-fried nuggets was significantly higher than oven-baked nuggets throughout the storage period. Ash and carbohydrate content were not significantly different (*p* > 0.05) among the two cooking methods at all time points. However, an increase in ash content was noticed with the increasing storage period, while the opposite was observed in the carbohydrate content, as shown in Table 1. The increase in protein, ash, and fat content during the storage period in deep-fried and oven-baked nuggets was associated with the reduction in moisture content during storage. Furthermore, changes in proximate composition are linked to mass balance as a result of moisture loss and oil uptake (in the case of frying), as well as other chemical changes such as thermal degradation of lipids. During cooking processes, a series of complex reactions such as denaturation, gelatinization, oxidation, hydrolysis, isomerization, and polymerization occur, influencing quality aspects such as flavor, texture, shelf life, and nutrient composition [26,29,30].

The energy values ranged from 243.3 to 277.4 and 207.2 to 233.3 kcal/100 g for deep-fried and oven-baked fish nuggets (Table 1), respectively. Bonfim et al. [18] also reported the energy values of 205.4 to 246.9 kcal/100 g for fried fish nuggets. The relatively high energy values in deep-fried nuggets are due to their significantly higher fat content.

### 3.2. Lipolysis and Lipid Oxidation

FFAs are the products of lipid hydrolysis. There was a general increase in the FFAs among the two cooking methods during storage (Figure 1a). The FFAs of the oven-baked fish nuggets were significantly higher than the deep-fried fish nuggets at all time points (Figure 1a). This was due to the longer heating time of the baking, which can cause more thermal degradation of lipids. Lower FFA values in deep-fried samples could also be owed to dilution caused by migration from the food’s interior into the cooking oil. Results revealed that there was a slight increase in FFAs throughout the storage period in both cooked samples (Figure 1a). This rise in FFAs might be because of incomplete inhibition of lipases/phospholipases during frozen storage and the thermal degradation of lipids during frying and baking of fish nuggets [14]. For deep-fried nuggets, the FFA content ranged from 1.02% to 1.46%, while that of the oven-baked nuggets was from 1.68% to 1.99%.

Figure 1b shows the changes in PV, an index of primary lipid oxidation product formation, of frozen cassava fish nuggets cooked by deep-frying and oven-baking. There was a significant difference in the PV of deep-fried and oven-baked fish nuggets during storage (*p* < 0.05). Results revealed that the PV of cooked samples increased with storage time, reaching 4.30 meq/kg fat in oven-baked nuggets and 4.03 meq/kg fat in deep-fried nuggets after 90 days. Findings from Moosavi-Nasab et al. [9] also demonstrated an increase in PV in fried fish nuggets, ranging from 1.54 to 4.34 meq/kg fat as the frozen storage period increased up to 90 days. The current study is consistent with previous studies that deep-fried nuggets had relatively lower PV than oven-baked nuggets. Results from Weber et al. [31] showed that fried catfish fillets had lower PV compared to baked samples. Several studies postulate that during deep-frying, the initial hydroperoxides decompose into various secondary oxidation products such as aldehydes, ketones, etc. [29,32,33,34]. During deep-fat frying, the hydroperoxides are often unstable. This could explain the method’s relatively low PV compared to oven-baking. Furthermore, the antioxidant present in the rice bran oil, e.g., γ-oryzanol and tocotrienols [35], may have helped to delay the oxidation of fried nuggets. As labeled, the rice bran oil used in this study contained approximately 8000 mg/kg of γ-oryzanol. It has been reported that frying with rice bran oil retards the lipid oxidation of fried dough during storage [35]. In addition, the baking time was longer than the frying time. Thus, the former had the higher PV than the later. Furthermore, as shown in Table 1, dilution owing to mass transfer of the frying oil into the food, as well as the fact that rice bran oil is more stable than fish oil [36], could all contribute to the lower PV of the deep-fried nuggets. PV is normally used to survey stability against oxidative rancidity [34]. A rancid flavor is perceptible at a PV of 10–20 meq/kg fat [37]. The lower values observed in the 90 days frozen storage implied that the cooked nuggets were within the recommended range for consumption. Furthermore, the ingredients such as the spices with the antioxidant potency may have shielded the fish nuggets from going rancid during the storage period [38].

### 3.3. Color

The color values of deep-fried and oven-baked cassava fish nuggets are illustrated in Table 2. Significant differences (*p* < 0.05) were observed for *L**, *a**, and *b** values during the storage among the treatments, indicating that the different cooking methods and storage periods affected the color parameters. The findings from this study showed that deep-fried nuggets tended to have higher *L**, *a**, and *b** values throughout the storage period. The *L** values of the nuggets cooked using both methods were positively correlated with their final moisture contents (R^2^ = 0.9294 for deep-fried nuggets and R^2^ = 0.9521 for oven-baked nuggets). Throughout the storage period, the *a** value of deep-fried nuggets was significantly higher than that of oven-baked nuggets (*p* < 0.05). The *b** values of deep-fried nuggets were likewise greater than those of oven-baked nuggets. As a result, the deep-fried nuggets were a vivid orange-yellow color, but the oven-baked nuggets were a pale yellow tone. Benjakul et al. [39] reported that nonenzymatic browning reactions, particularly the Maillard reaction and caramelization, generally occurred upon heating of food containing reducing sugars and amino acids. According to Zhang et al. [29], lipid oxidation and Maillard reaction are two of the most common and well-studied changes in both the frying oil and fried food. Hence, it may be postulated that frying generated more Maillard reaction products when compared to baking. Furthermore, the rice bran oil has a dark yellow color, and absorption could affect the color of the fried nuggets. According to Bechtel et al. [40], the most ideal color for fried products was a light brown color, depending on the food type. The golden color of fried crusts appeared due to the Maillard reaction on the surfaces of the fish fillets [41,42,43]. Thus, the application of deep-frying is among the factors responsible for color properties in fish nuggets.

With increasing storage time, all the color parameters of both treatments tended to decrease (Table 2). This may be due to the decreases in the substrates for nonenzymatic browning reaction, particularly carbohydrate [39,44], in both samples with increasing frozen storage time (Table 1).

### 3.4. TEF and TPA

The TEF of fish nuggets from the two cooking methods are shown in Table 3. The TEF ranged from 10.74–25.52% and 2.91–8.00% for deep-fried and oven-baked nuggets, respectively. In general, TEF can be used to estimate the fat and water holding efficacies of meat products [23,45]. Significant differences were observed among the two cooking methods (*p* < 0.05). In addition, the TEF of both samples increased with storage time (*p* < 0.05). The baked fish nuggets had relatively lower TEF than the deep-fried nuggets at all time points (*p* < 0.05), which is expected because the wet process (frying) will intuitively retain and express more fluid than the dry process (baking). The low fat content of oven-baked nuggets (Table 1) is responsible for their low TEF. In duck meat sausages supplemented with cereal flours, the lower the fat, the lower the TEF [46]. In comparison to baking, frying causes more moisture migration from the cooked nuggets. The loss of moisture from the fried nuggets causes oil uptake to fill the spaces previously occupied by water molecules (Table 1). Because the absorbed oil is not strongly connected to the nugget structures, more TEF can be detected at all time points. In fish-meat-based snacks, frying was shown to have less tight and immobilized water, whereas baking had a lot of bound water [43].

TPA results for deep-fried and oven-baked cassava fish nuggets were determined as hardness, springiness, cohesiveness, and chewiness (Table 3). There was a difference in textural properties among samples. According to Moyano et al. [47], textural changes during cooking are caused by a variety of physical, chemical, and structural changes. Principally, hardness reflects the force required to compress the food between the molars [48]. The hardness of deep-fried nuggets was higher than the baked nuggets throughout the storage period (*p* < 0.05). According to Bechtel et al. [40], baked breaded and battered catfish had a lower hardness value than fried breaded and battered catfish. The crust formation may be linked to the greater hardness value in deep-fried nuggets. Cooking foods causes starch gelatinization, protein denaturation, and water evaporation. Regardless of the cooking process used in this study, moisture content was negatively correlated with hardness, whereas protein and fat levels were positively correlated with hardness (R^2^ = 0.6203–0.7786). Moisture content was negatively correlated with hardness, whereas protein and fat levels were positively correlated with hardness. As a result, the final textural quality of the fish nuggets could be influenced by compositional variation. Springiness and cohesiveness of both samples showed similar degrees of change at different time points. Springiness represents the deformation when the compressive force is taken away [49]. Cohesiveness represents the resistance of the internal bonds [48]. Springiness of the deep-fried nuggets was higher than oven-baked nuggets (*p* < 0.05) whereas the cohesiveness values of both samples were not different (*p* > 0.05). Based on the findings, oven-baked fish nuggets were softer compared to the deep-fried nuggets. The chewiness had the same trend with the hardness and springiness. Chewiness is the derived textural parameter, and its behavior is governed by the primary parameters it is dependent on [1]. Chewiness reflects the energy needed to chew the solid food until it is ready for swallowing [49]. The chewiness of deep-fried nuggets was higher than baked nuggets until the end of the storage period (*p* < 0.05). It has been reported that the existence of crust impacts the food’s mechanical qualities, as well as its texture and acceptability [50]. As a result, the chewiness and hardness of deep-fried nuggets may be related to crust development. The chewiness of both nuggets changed in a similar way with the hardness, as it did in other products such as deep-fried battered and breaded fish balls [51] and gluten-free breads [52]. With increasing storage time, all textural parameters tended to increase in both samples suggesting the hardening of the nuggets with prolonged storage. This was probably due to the aggregation of proteins, the loss of water, and the oxidation/hydrolysis of lipid during frozen storage [53,54]. Those changes may influence the texture of the final cooked nuggets. It has been reported that FFA and aldehydic lipid oxidation products can cause the protein cross-linking [55] and, hence, muscle toughening during frozen storage and subsequent cooking of fish nuggets.

### 3.5. Microbiological Quality

The TPC and yeast and mold counts are shown in Figure 2. The initial TPC for deep-fried and oven-baked fish nuggets were 3.85 log CFU/g and 5.33 log CFU/g, respectively, and decreased to 1.33 log CFU/g and 2.50 log CFU/g, respectively, during 90 days of frozen storage (Figure 2a). The TPC was within the maximum of 6 log CFU/g set for cooked meat products [14,56]. From the results, oven-baked fish nuggets had a significantly higher TPC than deep-fried nuggets from Day 0 to Day 15 of storage (*p* < 0.05). Then, no significant differences were seen among the two cooking methods from Day 30 until the end (*p* > 0.05). However, there was a decrease in TPC in both treatments as the storage period increased (Figure 2a). The yeast and mold count of the fish nuggets decreased with the storage period (Figure 2b). However, numbers did not exceed the acceptable limit. It has been reported that the acceptability limit of yeast and mold for cooked meat was 2 log CFU/g [57]. Cross contamination can occur after cooking, leading to the discovery of microflora in the samples. As a result, the use of good manufacturing practices (GMP) and good hygiene practices (GHP) is recommended. The hygienic handling can lower the initial microbial load [9]. Furthermore, the reduction in the microbial count may be due to the effect of freezing [9,58]. Thus, the use of blast freezer to quickly freeze the fish nuggets before keeping in the freezer may be a contributing factor.

## 4. Conclusions

The cooking methods (deep-frying and oven-baking) and the frozen storage influenced the overall quality of the cassava fish nuggets. Proximate composition, FFA, PV, color, texture, and microbial quality of the nuggets varied among the two cooking methods and frozen storage time. In general, deep frying produced fish nuggets with a better quality, as evidenced by lower PV, FFA, TPC, and yeast and mold count compared to oven-baked nuggets during frozen storage. However, the oven-baked nuggets had the lower fat content and energy value and relatively higher moisture content. Fish nuggets can be kept at −18 °C for up to 90 days without marked deterioration. Microbial indices of fish nuggets evaluated during 90 days of frozen storage were also found to be within acceptable quality standards for cooked fish products. This study has demonstrated that deep-frying and oven-baking produced nuggets of high eating quality during frozen storage. However, for food safety concerns, it is suggested that GMP and GHP be used in the production of fish nuggets.

## Figures and Tables

**Figure 1 foods-10-03158-f001:**
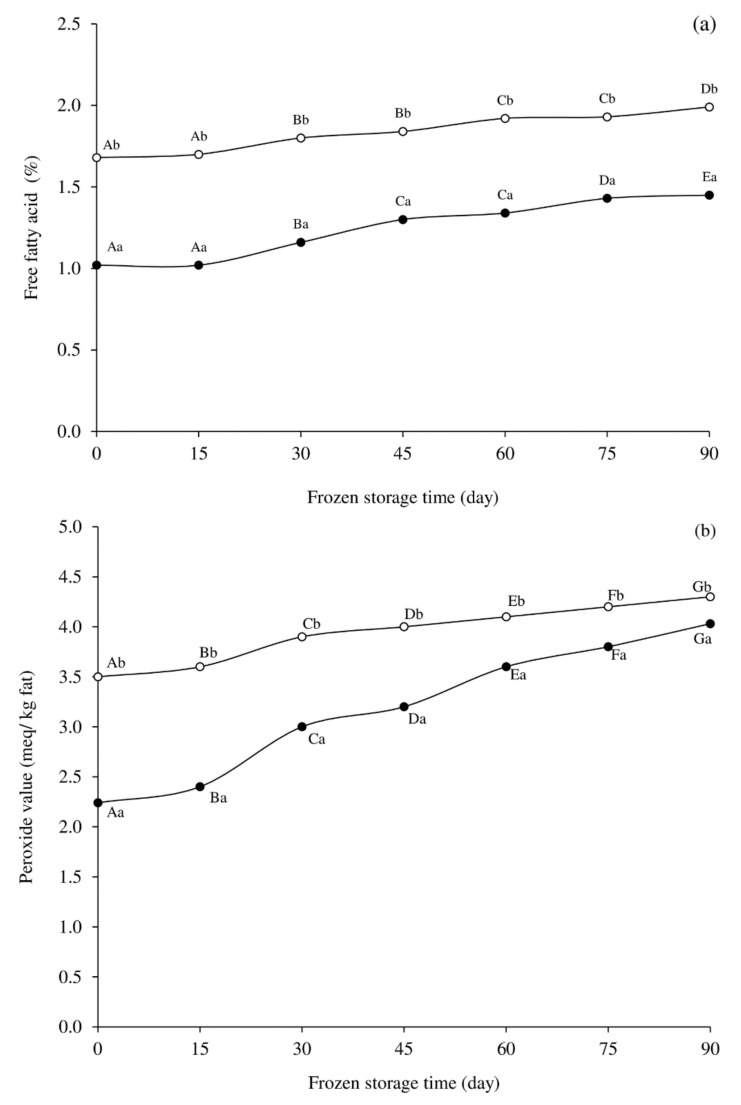
Changes in free fatty acids (**a**) and peroxide value (**b**) of frozen cassava fish nuggets cooked by deep-frying (●) and oven-baking (○). Different upper case letters in the same cooking method (between storage times) and different lower case letters between cooking methods at the same time indicate significant difference (*p* < 0.05).

**Figure 2 foods-10-03158-f002:**
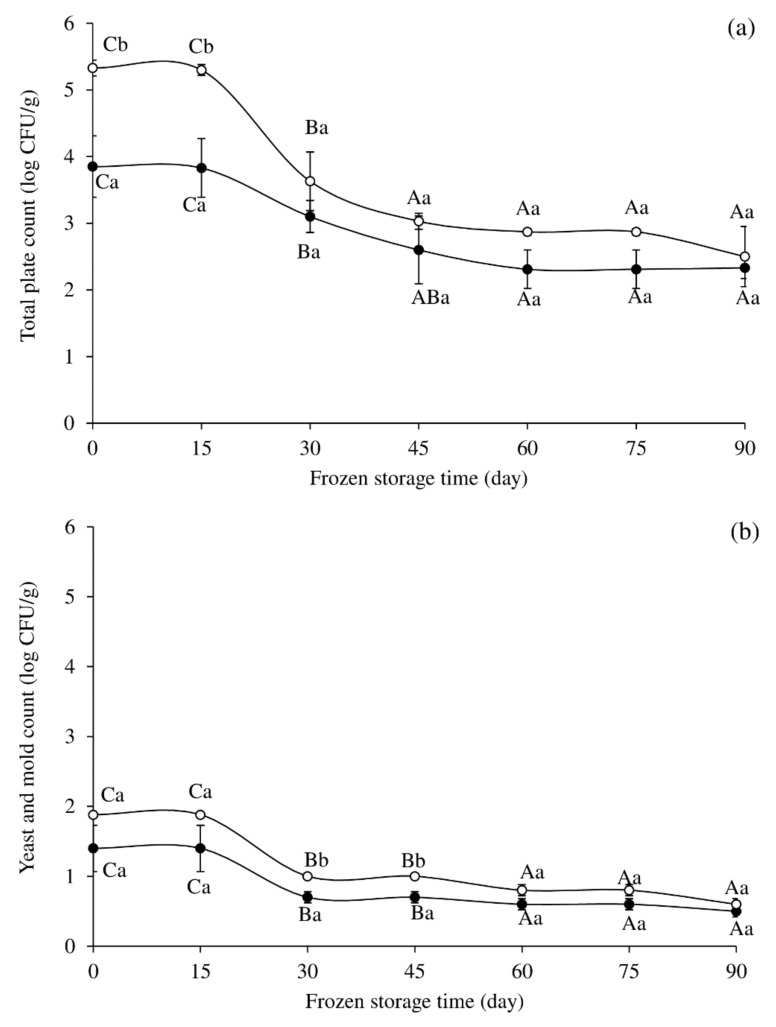
Changes in total plate count (**a**) and yeast and mold count (**b**) of frozen cassava fish nuggets cooked by deep-frying (●) and oven-baking (○). The bars indicate standard deviation from triplicate determinations. Different upper case letters in the same cooking method (between storage times) and different lower case letters between cooking methods at the same time indicate significant difference (*p* < 0.05).

**Table 1 foods-10-03158-t001:** Proximate composition and energy value of frozen cassava fish nuggets cooked by deep-frying and oven-baking.

Composition	Samples	Frozen Storage Time (Day)						
		0	15	30	45	60	75	90
Moisture (%)	Deep-fried	49.6 ± 0.3 ^aC^	49.6 ± 0.9 ^aC^	47.3 ± 0.2 ^aB^	47.4 ± 0.4 ^aB^	46.6 ± 0.3 ^aAB^	46.0 ± 0.1 ^aA^	45.8 ± 0.7 ^aA^
	Oven-baked	53.1 ± 0.5 ^bC^	52.5 ± 0.7 ^bC^	51.2 ± 0.7 ^bB^	50.5 ± 0.3 ^bB^	49.5 ± 0.3 ^bAB^	49.3 ± 0.1 ^bA^	49.1 ± 0.1 ^bA^
Protein (%)	Deep-fried	33.0 ± 1.1 ^aA^	32.2 ± 1.3 ^aA^	33.7 ± 0.2 ^aA^	34.0 ± 0.1 ^aB^	34.6 ± 0.3 ^aC^	35.0 ± 0.1 ^aC^	35.2 ± 0.2 ^aC^
	Oven-baked	33.7 ± 0.5 ^aA^	34.6 ± 0.5 ^aAB^	35.5 ± 0.6 ^bAB^	36.8 ± 0.3 ^bB^	36.4 ± 0.5 ^bB^	37.6 ± 0.1 ^bC^	37.7 ± 0.4 ^bC^
Fat (%)	Deep-fried	11.1 ± 0.9 ^bA^	11.1 ± 0.9 ^bA^	12.4 ± 0.6 ^bAB^	13.2 ± 0.2 ^bAB^	13.8 ± 0.2 ^bB^	14.3 ± 0.3 ^bB^	14.7 ± 0.2 ^bB^
	Oven-baked	6.2 ± 0.5 ^aA^	6.2 ± 0.5 ^aA^	7.4 ± 0.2 ^aB^	7.6 ± 0.3 ^aB^	7.9 ± 0.2 ^aB^	8.4 ± 0.2 ^aC^	8.9 ± 0.2 ^aC^
Ash (%)	Deep-fried	2.6 ± 0.7 ^aA^	2.6 ± 0.7 ^aA^	2.8 ± 0.1 ^aA^	2.9 ± 0.0 ^aA^	2.9 ± 0.1 ^aAB^	3.0 ± 0.0 ^aAB^	3.2 ± 0.2 ^aB^
	Oven-baked	2.8 ± 0.0 ^aA^	2.8 ± 0.0 ^aA^	2.8 ± 0.2 ^aAB^	3.0 ± 0.0 ^aB^	3.0 ± 0.1 ^aB^	3.0 ± 0.0 ^aB^	3.6 ± 0.2 ^aC^
Carbohydrate (%)	Deep-fried	3.8 ± 0.3 ^aC^	3.8 ± 0.3 ^aC^	3.8 ± 0.4 ^aC^	2.4 ± 0.3 ^aAB^	2.1 ± 0.5 ^aAB^	1.7 ± 0.2 ^aA^	1.1 ± 1.0 ^aA^
	Oven-baked	4.2 ± 0.6 ^aD^	4.2 ± 0.6 ^aD^	3.0 ± 0.4 ^aC^	2.1 ± 0.3 ^aB^	3.2 ± 0.4 ^aC^	0.9 ± 0.6 ^aA^	0.7 ± 0.2 ^aA^
Energy (kcal/100 g)	Deep-fried	246.9 ± 3.8 ^bA^	243.3 ± 1.8 ^bA^	261.8 ± 2.0 ^bB^	264.9 ± 1.9 ^bB^	270.9 ± 1.9 ^bC^	275.8 ± 2.3 ^bD^	277.4 ± 3.2 ^bD^
	Oven-baked	207.2 ± 4.7 ^aA^	210.7 ± 4.2 ^aA^	220.9 ± 3.4 ^aB^	224.0 ± 1.6 ^aB^	229.1 ± 0.7 ^aC^	229.4 ± 1.0 ^aC^	233.3 ± 1.8 ^aD^

Values are mean ± standard deviation from triplicate determinations. Different upper case letters in the same row (between storage times) and different lower case letters among samples (between cooking methods) at the same time indicate significant difference (*p* < 0.05).

**Table 2 foods-10-03158-t002:** Color of frozen cassava fish nuggets cooked by deep-frying and oven-baking.

Color	Samples	Frozen Storage Time (Day)						
		0	15	30	45	60	75	90
*L**	Deep-fried	79.63 ± 0.42 ^bD^	79.38 ± 0.71 ^bD^	77.19 ± 0.24 ^bC^	75.87 ± 0.51 ^aB^	76.41 ± 0.43 ^bB^	75.18 ± 0.06 ^bA^	75.17 ± 0.13 ^bA^
	Oven-baked	77.32 ± 0.16 ^aE^	77.28 ± 0.13 ^aE^	76.24 ± 0.17 ^aD^	75.61 ± 0.43 ^aC^	75.23 ± 0.17 ^aC^	74.42 ± 0.02 ^aB^	74.15 ± 0.11 ^aA^
*a**	Deep-fried	10.26 ± 0.18 ^bC^	10.20 ± 0.16 ^bC^	9.75 ± 0.11 ^bB^	9.55 ± 0.27 ^bAB^	9.83 ± 0.12 ^bB^	9.76 ± 0.02 ^bB^	9.47 ± 0.01 ^bA^
	Oven-baked	5.06 ± 0.09 ^aF^	5.00 ± 0.15 ^aF^	4.81 ± 0.02 ^aE^	4.43 ± 0.16 ^aD^	3.98 ± 0.02 ^aC^	3.45 ± 0.00 ^aB^	3.07 ± 0.08 ^aA^
*b**	Deep-fried	28.18 ± 0.13 ^bD^	28.17 ± 0.12 ^bD^	28.00 ± 0.16 ^bD^	27.31 ± 0.22 ^aC^	27.53 ± 0.25 ^bC^	26.46 ± 0.09 ^aB^	25.20 ± 0.16 ^bA^
	Oven-baked	27.48 ± 0.02 ^aD^	27.48 ± 0.01 ^aD^	27.06 ± 0.09 ^aC^	26.43 ± 0.41 ^aB^	26.14 ± 0.12 ^aB^	26.41 ± 0.01 ^aB^	24.20 ± 0.16 ^aA^

Values are mean ± standard deviation from triplicate determinations. Different upper case letters in the same row (between storage times) and different lower case letters among samples (between cooking methods) at the same time indicate significant difference (*p* < 0.05).

**Table 3 foods-10-03158-t003:** Total expressible fluid (TEF) and texture profile analysis (TPA) of frozen cassava fish nuggets cooked by deep-frying and oven-baking.

Parameters	Samples	Frozen Storage Time (Day)						
		0	15	30	45	60	75	90
TEF (%)	Deep-fried	10.74 ± 0.10 ^bA^	13.86 ± 0.16 ^bB^	14.64 ± 0.11 ^bC^	14.84 ± 0.20 ^bC^	15.84 ± 0.06 ^bD^	20.92 ± 0.15 ^bE^	25.52 ± 0.50 ^bF^
	Oven-baked	2.91 ± 0.12 ^aA^	3.51 ± 0.14 ^aB^	4.77 ± 0.16 ^aC^	5.90 ± 0.16 ^aD^	6.87 ± 0.10 ^aE^	8.00 ± 0.20 ^aF^	7.42 ± 0.60 ^aF^
TPA								
Hardness (g)	Deep-fried	5627 ± 0.2 ^bA^	5629 ± 0.3 ^bB^	5632 ± 1 ^bC^	5726 ± 1 ^bD^	5730 ± 25 ^bDE^	5773 ± 19 ^bE^	5705 ± 6 ^bE^
	Oven-baked	4624 ± 0.3 ^aBC^	4606 ± 2 ^aA^	4625 ± 1 ^aBC^	4633 ± 3 ^aC^	4725 ± 4 ^aBC^	4729 ± 3 ^aBC^	4765 ± 3 ^aD^
Springiness (cm)	Deep-fried	0.78 ± 0.00 ^bA^	0.83 ± 0.01 ^bB^	0.85 ± 0.00 ^bC^	0.95 ± 0.02 ^bE^	0.95 ± 0.01 ^bE^	0.92 ± 0.01 ^bD^	0.92 ± 0.01 ^bD^
	Oven-baked	0.75 ± 0.25 ^aABC^	0.74 ± 0.01 ^aB^	0.73 ± 0.01 ^aB^	0.63 ± 0.12 ^aA^	0.86 ± 0.04 ^aC^	0.86 ± 0.03 ^aC^	0.88 ± 0.01 ^aC^
Cohesiveness	Deep-fried	0.62 ± 0.00 ^aA^	0.61 ± 0.01 ^aA^	0.68 ± 0.00 ^aB^	0.79 ± 0.08 ^aCD^	0.72 ± 0.03 ^aC^	0.85 ± 0.04 ^aD^	0.63 ± 0.12 ^aAB^
	Oven-baked	0.64 ± 0.21 ^aBC^	0.57 ± 0.03 ^aA^	0.64 ± 0.04 ^aAB^	0.81 ± 0.08 ^aC^	0.81 ± 0.06 ^aC^	0.82 ± 0.04 ^aC^	0.80 ± 0.08 ^aC^
Chewiness (g·cm)	Deep-fried	2024 ± 0.2 ^bA^	2084 ± 0.3 ^bC^	2076 ± 4 ^bB^	2091 ± 1 ^bD^	2093 ± 3 ^bD^	2139 ± 4 ^bE^	2145 ± 4 ^bE^
	Oven-baked	1324 ± 7 ^aB^	1311 ± 0.2 ^aA^	1326 ± 1 ^aB^	1338 ± 3 ^aC^	1350 ± 3 ^aD^	1354 ± 6 ^aD^	1349 ± 1 ^aD^

Values are mean ± standard deviation from triplicate determinations. Different upper case letters in the same row (between storage times) and different lower case letters among samples (between cooking methods) at the same time indicate significant difference (*p* < 0.05).

## Data Availability

The datasets generated for this study are available on request to the corresponding author.

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
