# Peer review of "Comparative Effect of Frying and Baking on Chemical, Physical, and Microbiological Characteristics of Frozen Fish Nuggets"

_foods, 2021, doi:10.3390/foods10123158_

Round 1

Reviewer 1 Report

The authors show the influence of two cooking methods (deep-frying and oven-baking) on the proximate composition, lipolysis, lipid oxidation, color, texture and microbial quality of cassava croaker (Pseudotolithus senegalensis) nuggets.

The aim of the study was to investigate the effects of deep-frying and oven-baking on the quality characteristics and technological properties 60 of frozen (-18 C/90 days) cassava fish nuggets.

The results provide a weak advance in current knowledge on the topic. Unfortunately, some hypotheses  are non adequate or they are not substantiated by references. The manuscript needs further References, as described below.

The study would be interesting from a practical point of view,but some modifications shall be considered. In some cases the analyses are not scientifically sound. In other  cases the Results  are not followed by a sufficient  discussion; the latter is not sometimes reported.

I will summarize the major issues identified as follows:

75: the weight of fish pieces is not sufficient; it would have been useful to specify the length, width, thickness of fish squares

79: the authors stated that “The nuggets were formulated using the recommended methods approved by Duman and Kuzgun [12] and Bechtel et al. [13] with slight modifications”.

The applied modifications don’t seem “slight”; in fact, substantial changes were applied. In my opinion it would be better if the authors described in detail  their methods.

In fact, in Duman and Kuzgun (2018) and Bechtel et al. (2018) the formulation of  fish nuggets was different if compared with the products described in this paper. In fact, fish fillets were mixed with different seasoning and ingredients. Duman and Kuzgun (2018) prepared also smoked meat nuggets .

Line 85: the authors report the proximate composition of raw fish nuggets. The preliminary samples preparation and the analytic methods are not specified. It is necessary to describe the methods or refer to any official methods of analysis. In any case, this part should be put before “nuggets preparation”.

Line 94: thawing method that was applied to the nuggets is not completely described. Thawing was carried out at 18-20°C, but the athors don’t write how long. Then, were the samples completely or partially thawed? It would be better if the authors wrote these indications.

Line 97: 2.4. Deep-frying and Oven-baking of Nuggets – Why the authors do not explain why they did they use different times of cooking for both types of nuggets?

Line 103: the authors should not report reference [13] but “Barbut, S. Producing Battered and Breaded Meat Products; American Meat Science Association: Champaign, IL, USA, 2012

From line 126 to Line 129: The authors used the method found in Panpipat and Chaijan (2017).  These authors considered that the procedure was “empirical”.   Thus it follows that the method is devoid of scientific rigor.

Line 151-152: the authors state that “In fresh nuggets (Day 0), a higher moisture content was recorded in oven-baked  nugget whereas a higher fat content was found in deep-fried nugget (p < 0.05) ” .  Rephrase the sentence, please

The obtained results were similar to those reported by Marimuthu et al. It should be better if the authors wrote this considerations and the exact citation (22) to support the concept.

Lines 152-154: the hypothesis reported in these lines is found only in Duan et al (21). The authors should modify the citations at the end of the sentence (line 154)

Line 158-159: the auhors should  give an explanation for the  moisture loss during frozen storage prior to cook the nuggets samples. The moisture loss was observed in both deep-fried and oven-baked nuggets; the authors should also specify this observation.

Line 159-161: Sentence ” At the same time….. of the cooked nuggets”: citation is required to support the concept

From Line 161 to line 170: the authors report the quantitative changes of some components but they do not make any assumption about the observed changes. They should discuss the results.

From line 218 to line 222 and line 245: citation is required to support the concept

Lines 252:  please, check the paper of Lowry and Tinsley, 1976 (reference 19); where did the authors  find the relationship between TEF and “water holding efficacies” and betweee TEF and fat estimation ? TEF can be useful to express emulsion stability (Panpipat and Chaijan, 2017; reference 20)

TEF can be useful  to express emulsion stability  in a food product (Reference 20 – Panpipat and Chaijan, 2017)

Line 255: citation is required to support the concept. I suggest that thawing time could have influenced the fluid retention capacity of the fish nuggets.; but the authors did not report the thawing time for nuggets.

Line 257: citation is required to support the concept at the end of the sentence

Lines 259-261: Water Holding Capacity (WHC) was not determined; on which basis the authors did the authors write the sentence in lines 259-260-261?

Consequently, the contents from  line 262 to line 265 should be refrased

Lines 271-272:it is not possible to make a statistic correlation between TEF and nuggets hardness; the authors’ statement is not supported by statistical analysis. The scientific assumption could be accepted.

Lines 277-278: the correlation analysis seems to refer to another study; please, change the sentence

From line 283 to line 294: the authors report the results but they do not  formulate a sufficient discussion

Line 287: the authors state that “The cohesiveness values of both samples were not different at all-time points”; but in lines 292-293 they report that “With increasing storage time, all textural parameters tended to increase in both samples”. It is necessary to change the first sentence.

Line 288: the authors write “structural integrity of the muscle” that “is intact since the fillets were not minced”. So, for example, they do not consider the influence of nuggets preparation and/or the interaction between samples preparation and cooking methods

Line 296: at the end of the sentence (i.e. “……frozen storage” citation is required to support the concept)

Paragraph “Microbial quality”: low TPC and low yeasts - mold load must be considered obvious for cooked products. Both cooking methods should destroy the microbial flora in the raw nuggets. The latter can be contaminated with the ingredients and/or seasonings and /or during the phases of preparation.

After cooking, a cross contamination could explain the reason why the authors have detected the microflora in the samples.

Anyway, in my opinion, microbial count should be necessary to evaluate the application of Good Manufacturing Pratices (GMP) and Good Hygiene Practices (GHP).

Line 335-336: “The investigation further uncovered that microbial indices of fish nuggets analyzed during 90 days frozen storage were within the worthy qualities for cooked fish items”

Frozen storage has a key role in the reduction of microflora; the latter is not destroyed , but low temperature, for a long time, can strongly inhibit microbial  growth. Once examined, the nuggets have a very low microbial load.  In any case this load depends on the level of contanimation found in raw samples. The latter  were not examined in tis study; thus, it is not possible to make a comparison with cooked nuggets.

Author Response

Reviewer 1

The authors show the influence of two cooking methods (deep-frying and oven-baking) on the proximate composition, lipolysis, lipid oxidation, color, texture and microbial quality of cassava croaker (Pseudotolithus senegalensis) nuggets.

The aim of the study was to investigate the effects of deep-frying and oven-baking on the quality characteristics and technological properties 60 of frozen (-18 °C/90 days) cassava fish nuggets.

The results provide a weak advance in current knowledge on the topic. Unfortunately, some hypotheses  are non adequate or they are not substantiated by references. The manuscript needs further References, as described below.

The study would be interesting from a practical point of view,but some modifications shall be considered. In some cases the analyses are not scientifically sound. In other  cases the Results  are not followed by a sufficient  discussion; the latter is not sometimes reported.

Ans: Thank you very much. A revision was carefully made accordingly.

I will summarize the major issues identified as follows:

75: the weight of fish pieces is not sufficient; it would have been useful to specify the length, width, thickness of fish squares

Ans: The length, width, and thickness of fish squares were given.

79: the authors stated that “The nuggets were formulated using the recommended methods approved by Duman and Kuzgun [12] and Bechtel et al. [13] with slight modifications”.

The applied modifications don’t seem “slight”; in fact, substantial changes were applied. In my opinion it would be better if the authors described in detail  their methods.

In fact, in Duman and Kuzgun (2018) and Bechtel et al. (2018) the formulation of  fish nuggets was different if compared with the products described in this paper. In fact, fish fillets were mixed with different seasoning and ingredients. Duman and Kuzgun (2018) prepared also smoked meat nuggets .

Ans: Thank you so much. Because of the diverse technique and raw materials used in the production of fish nuggets. Those references were removed, and we went over our approaches in detail.

Line 85: the authors report the proximate composition of raw fish nuggets. The preliminary samples preparation and the analytic methods are not specified. It is necessary to describe the methods or refer to any official methods of analysis. In any case, this part should be put before “nuggets preparation”.

Ans: Because the nuggets have to be prepared first, so, we stated that “According to AOAC [15], the proximate composition of the raw nuggets was 55.3±0.9% moisture, 34.0±0.1% protein, 5.9±0.1% fat, 2.5±0.1% ash, and 2.3±0.8% carbohydrate.

Line 94: thawing method that was applied to the nuggets is not completely described. Thawing was carried out at 18-20°C, but the athors don’t write how long. Then, were the samples completely or partially thawed? It would be better if the authors wrote these indications.

Ans: The detail for thawing was rechecked and stated that “Frozen samples were taken for deep-frying or oven-baking, after partially thawing for 60 min at 4 °C to a core temperature of -7 to -5 °C, at 15 days intervals and further analyses were carried out.

Line 97: 2.4. Deep-frying and Oven-baking of Nuggets – Why the authors do not explain why they did they use different times of cooking for both types of nuggets?

Ans: It was stated that “Due to the differences in heating medium, different cooking times were used for both types of nuggets in order to achieve the desired quality in the finished products.

Line 103: the authors should not report reference [13] but “Barbut, S. Producing Battered and Breaded Meat Products; American Meat Science Association: Champaign, IL, USA, 2012

Ans: Done.

From line 126 to Line 129: The authors used the method found in Panpipat and Chaijan (2017).  These authors considered that the procedure was “empirical”.   Thus it follows that the method is devoid of scientific rigor.

Ans: The original method was cited.

Line 151-152: the authors state that “In fresh nuggets (Day 0), a higher moisture content was recorded in oven-baked  nugget whereas a higher fat content was found in deep-fried nugget (p < 0.05) ” .  Rephrase the sentence, please

The obtained results were similar to those reported by Marimuthu et al. It should be better if the authors wrote this considerations and the exact citation (22) to support the concept.

Ans: It was changed to “In fresh nuggets (Day 0), a higher moisture content was found in oven-baked nugget whereas a higher fat content was found in deep-fried nugget (p < 0.05). The results obtained were comparable to those published by Marimuthu et al. [24].

Lines 152-154: the hypothesis reported in these lines is found only in Duan et al (21). The authors should modify the citations at the end of the sentence (line 154)

Ans: According to the recommendation of Reviewer#2, the statement was changed to “During deep-frying, water was evaporated and frying oil was undergone diffusion to the fish flesh, leading to a higher fat content in deep-fried fish than in oven-baked fish [24, 25]. The oil uptake is largely by absorption via porous pores and capillaries. Parkash Kochhar and Gertz [26] proposed that the mechanism of oil absorption during deep frying is dependent on the amount of water removed and the way in which this moisture is lost. Once the fried product is taken from the hot oil and begins to cool, oil uptake is primarily a surface-related phenomenon caused by a competition between drainage and suction into the porous crust [27].

Line 158-159: the auhors should  give an explanation for the  moisture loss during frozen storage prior to cook the nuggets samples. The moisture loss was observed in both deep-fried and oven-baked nuggets; the authors should also specify this observation.

Ans: It was changed to “Moisture loss, due to dehydration, during frozen storage may explain the decrease in moisture content of the cooked nuggets.

Line 159-161: Sentence ” At the same time….. of the cooked nuggets”: citation is required to support the concept

Ans: A reference was cited. “At the same time, the loss of breadcrumbs from the coating during frozen storage and cooking may cause the decrease in carbohydrate content of the cooked nuggets. Loss of coating can occur at any stage during the preparation or frozen storage [28].

From Line 161 to line 170: the authors report the quantitative changes of some components but they do not make any assumption about the observed changes. They should discuss the results.

Ans: The discussion was given. “The protein content was higher in oven-baked nuggets than in deep-fried nuggets from Day 15 to Day 90 of storage. The fat content in deep-fried nuggets was significantly higher than oven-baked nuggets throughout the storage period. Ash and carbohydrate content were not significantly different (p > 0.05) among the two cooking methods at all-time points. However, an increase in ash content was noticed with the increasing storage period whiles a reverse was observed in the carbohydrate content as shown in Table 1. The increase in protein, ash, and fat content during the storage period in deep-fried and oven-baked nuggets was associated with the reduction in moisture content during storage. Furthermore, changes in proximate composition are linked to mass balance as a result of moisture loss and oil uptake (in the case of frying), as well as other chemical changes such as lipid thermal degradation. During the cooking process, a series of complicated processes such as denaturation, gelatinization, oxidation, hydrolysis, isomerization, and polymerization occur, influencing quality aspects such as flavor, texture, shelf life, and nutrient composition [26, 29, 30].

From line 218 to line 222 and line 245: citation is required to support the concept

Ans: References were added.

Lines 252:  please, check the paper of Lowry and Tinsley, 1976 (reference 19); where did the authors  find the relationship between TEF and “water holding efficacies” and betweee TEF and fat estimation ? TEF can be useful to express emulsion stability (Panpipat and Chaijan, 2017; reference 20)

TEF can be useful  to express emulsion stability  in a food product (Reference 20 – Panpipat and Chaijan, 2017)

Ans: It is our mistake. Actually, it was Panpipat and Chaijan (2017). Also, it was already stated that “In general, TEF can be used to estimate the fat and water holding efficacies of meat product [23, 45].”

Line 255: citation is required to support the concept. I suggest that thawing time could have influenced the fluid retention capacity of the fish nuggets.; but the authors did not report the thawing time for nuggets.

Ans: The thawing time was reported in ‘Section 2.3. Nugget Preparation from Fish Fillet.’ The discussion was revised. “In addition, the TEF of both samples increased as storage time increased (p < 0.05). Therefore, the cooking methods and frozen storage periods affected the fluid retention capacity of the fish nuggets. The baked fish nuggets had relatively lower TEF than the deep-fried nuggets at all-time points (p < 0.05), which is expected because the wet process (frying) will intuitively retain and express more fluid than the dry process (baking). The low-fat content of oven-baked nuggets (Table 1) is responsible for its low TEF. In duck meat sausages supplemented with cereal flours, the lower the fat, the lower the TEF [46]. In comparison to baking, frying caused more moisture migration. More importantly, moisture migration causes oil uptake to fill the spaces previously occupied by water molecules (Table 1). Because the adsorbed oil is not strongly connected to the nugget structures, more TEF can be detected at all-time points. In fish meat-based snacks, frying was shown to have less tight and immobilized water, whereas baking had a lot of tight and immobilized water [43].”

Line 257: citation is required to support the concept at the end of the sentence

Ans: Citation was added.

Lines 259-261: Water Holding Capacity (WHC) was not determined; on which basis the authors did the authors write the sentence in lines 259-260-261?

Consequently, the contents from  line 262 to line 265 should be refrased

Ans: This assumption was deleted and the new statement was added as suggested by Reviewer #2. “In comparison to baking, frying caused more moisture migration. More importantly, moisture migration causes oil uptake to fill the spaces previously occupied by water molecules (Table 1). Because the adsorbed oil is not strongly connected to the nugget structures, more TEF can be detected at all-time points. In fish meat-based snacks, fry-ing was shown to have less tight and immobilized water, whereas baking had a lot of tight and immobilized water [43].

Lines 271-272:it is not possible to make a statistic correlation between TEF and nuggets hardness; the authors’ statement is not supported by statistical analysis. The scientific assumption could be accepted.

Ans: It was changed to “The hardness of deep-fried nuggets was higher than baked nuggets till the end of the storage period (p < 0.05). According to Bechtel et al. [41], baked breaded and battered catfish had a lower hardness value than fried breaded and battered catfish. The crust formation may be linked to the greater hardness value in deep-fried nuggets. Cooking foods causes starch gelatinization, protein denaturation and water evaporation.

Lines 277-278: the correlation analysis seems to refer to another study; please, change the sentence

Ans: It was calculated from this study. So, it was changed to “Regardless of the cooking process used in this investigation, a correlation analysis be-tween hardness and major composition (moisture, protein, and fat) of the fish nuggets revealed that hardness is linked to those components (R2 = 0.6203-0.7786). Moisture content was negatively correlated with hardness, whereas protein and fat levels were positively correlated with hardness. As a result, the final textural quality of the fish nuggets could be influenced by compositional variation.

From line 283 to line 294: the authors report the results but they do not  formulate a sufficient discussion

Ans: A discussion was revised as recommended. “Springiness represents the deformation when the compressive force is taken away [49]. Cohesiveness represents the resistance of the internal bonds [48]. Springiness of the deep-fried nuggets was higher than oven-baked nuggets (p < 0.05) whereas the cohesiveness values of both samples were not different (p > 0.05). Based on the findings, oven-baked fish nuggets were softer compared to the deep-fried nuggets. The chewiness had the same trend with the hardness and springiness. Chewiness is the derived textural parameter, and its behavior is governed by the primary parameters it is dependent on [1]. Chewiness reflects the energy needed to chew the solid food until it is ready for swallowing [49]. The chewiness of deep-fried nuggets was higher than baked nuggets till the end of the storage period (p < 0.05). This textural property may be linked to the production of crust in deep-fried nuggets. It has been reported that the existence of crust impacts the food's mechanical qualities, as well as its texture and acceptability [50]. The chewiness of both nuggets changed in a similar way with the hardness, as it did in other products such deep-fried battered and breaded fish balls [51] and gluten-free breads [52]. With increasing storage time, all textural parameters tended to increase in both samples suggesting the hardening of the nuggets with prolonged storage.

Line 287: the authors state that “The cohesiveness values of both samples were not different at all-time points”; but in lines 292-293 they report that “With increasing storage time, all textural parameters tended to increase in both samples”. It is necessary to change the first sentence.

Ans: Done.

Line 288: the authors write “structural integrity of the muscle” that “is intact since the fillets were not minced”. So, for example, they do not consider the influence of nuggets preparation and/or the interaction between samples preparation and cooking methods

Ans: The discussion about the structural integrity of the muscle was deleted and a new statement was given.

Line 296: at the end of the sentence (i.e. “……frozen storage” citation is required to support the concept)

Ans: Done.

Paragraph “Microbial quality”: low TPC and low yeasts - mold load must be considered obvious for cooked products. Both cooking methods should destroy the microbial flora in the raw nuggets. The latter can be contaminated with the ingredients and/or seasonings and /or during the phases of preparation.

After cooking, a cross contamination could explain the reason why the authors have detected the microflora in the samples.

Anyway, in my opinion, microbial count should be necessary to evaluate the application of Good Manufacturing Pratices (GMP) and Good Hygiene Practices (GHP).

Ans: Thank you very much for your invaluable suggestion. The statement was added “Cross contamination can occur after cooking, leading to the discovery of microflora in the samples. As a result, the use of Good Manufacturing Practices (GMP) and Good Hygiene Practices (GHP) is recommended.

Line 335-336: “The investigation further uncovered that microbial indices of fish nuggets analyzed during 90 days frozen storage were within the worthy qualities for cooked fish items”

Frozen storage has a key role in the reduction of microflora; the latter is not destroyed , but low temperature, for a long time, can strongly inhibit microbial  growth. Once examined, the nuggets have a very low microbial load.  In any case this load depends on the level of contanimation found in raw samples. The latter  were not examined in tis study; thus, it is not possible to make a comparison with cooked nuggets.

Ans: The ingredients used to make fish nuggets all come from the same source. The only variation is the method of cooking, which is either baking or deep-frying. As a result, the microbial growth could be due to frozen storage and cross contamination, as suggested. The statement was changed to “Microbial indices of fish nuggets evaluated during 90 days of frozen storage were also found to be within acceptable quality standards for cooked fish products. This study has demonstrated that deep-frying and oven-baking produced nuggets of high eating quality during frozen storage. However, for food safety concerns, it is suggested that GMP and GHP be used in the production of fish nuggets.

Reviewer 2 Report

The manuscript reports on the ‘comparative effect of frying and baking on the chemical, physical and microbiological characteristics of frozen fish nuggets. There are relatively similar published works exploring similar themes – cooking effects on fish nuggets.  However, the main variances here is the geographical setting and the relevance to the target, and species of fish used for the study which lends to value-addition to this fish product – contributing to postharvest strategies. Overall, the conclusions are consistent with the results and discussions. Authors should consider adding a contextual relevance of cassava croaker to the geographical location in the introduction.

Furthermore, aspects of the manuscript require further clarification and correction. For example, the idea that during frying, the mass transfer of oil into the fried food is by ‘osmosis’ – which is inaccurate as osmosis requires a selective semi-preamble membrane. Likewise, of the explanations were speculative without available evidence to support them.

Specific comments are listed below.

Line 14: Revise to " The moisture, protein, fat, ash, and carbohydrate contents varied ...."

Line 15-16: This is expected considering the cooking method was frying with oil

Lines 27-28, This statement is not clear. Please revise. The first part is fine but the second part starting with food products with identified ...." is not clear and reads incomplete

Lines 29-30: What does 'Unique chemical" properties mean? Fish has always been a protein source, hence, what makes that unique – Please include a reference for this statement - to support "promote consumer health and prevent some diseases"

Lines 32-33: What’s the meaning of 'creature protein'? Are they sources of supplements or are supplements? It should be the former - please review this statement

Lines 35-36: It is either a source or it isn't. Does it contain those minerals? If yes, then remove 'possible'

Lines 37-38: Consider this revision "Thus, this fish has good nutritive values and can contribute positively to human nutrition"

Line 50: Is 'food nutrition' a correct phrase?

Line 53: ...are regarded as two of the oldest food processing methods

Lines 54-56: Please rephrase this to remove the references to 'field of sustenance' as the statement is well known in food processing, nutrition, and physiology. Basically, 'The type of cooking processing can damage important nutrients in foods.

Line 60: Why technological properties? In food processing, technological properties refer to things such as emulsification, foaming properties etc.

The current study is largely about quality indices - physical, chemical and textural properties. Please revise

Lines 66-67: Is it required to list all these analytical chemicals? Or best to state that "All analytical chemicals were purchased from Sigma-Aldrich (St. Louis, MO, USA)?

Lines 70-72: Consider this revision "Cassava croaker (P. senegalensis) weighing 1-2 kg were purchased from a local market in James Town,

Greater Accra, Ghana, West Africa." Remove 'along the coast ..."

Lines 72-73: What does 'off stacked' mean? Also, was it only one fish? Ideally, multiple fishes are a better sample representation.

... brought to the laboratories of ...

Please clarify this - If fish was purchased from a local supermarket, then clarify the statement in lines 72-73 - The fish was off stacked........

Should be 'a' local supermarket ...

Line 75: ... gutted, cleaned, and filleted. The fish fillets were ......

Line 78: ...For each treatment, approximately 182 fillets ......

Please clarify this - You produced fillets that were cut into 17-18g square cubes/shapes - then they cease to be 'fillets'. Hence should the statement be "... approximately, 182 of cut fillet cubes were used.' There is no need to repeat the 17-18 g (already stated in line 75). Alternatively, give the cut fillets a name that you may refer to, e.g., product A.

Line 79: Delete 'approved'. How did these authors approve of such a method? An industry body or government agency has that responsibility.

Use '... using the method recommended by Duman and Kuzgun, and ...

Line 80: Note that 'you are using the cut cubes not the fillets - Fillets are thin cuts lengthwise along the bone, however, they have been cut into cubes, hence, no longer fillets

Lines 81-82: All these ingredients are seasonings, hence clarify '27.8 f of complete seasoning' - what is it? On-the market ready-made composite seasoning ingredient? maybe include some details - brand/product name. Also, is it 15.5 g of lemon juice? or the whole lemon? Please indicate which part.

Line 82: Please review the appropriateness of the use of the term 'fillet' in different sections of the manuscript - I think a better classification/term will help clarify the process better - that is - you start with the harvested fish - make fillet - cut into small pieces (dicing) - then mixing and seasoning etc.

Lines 85-86: Suggestion, 'The proximate composition of the uncooked fish nugget was 55.3 ....'

Should the uncooked be 'raw nuggets'? As they are intuitively the same

Line 85: Revise - 'The nuggets were kept on a tray for 30 min ..."

Line 88:  ... were either oven-baked or deep-fried ... Using 'add' suggests the use baked and fried the same product

Lines 92: Should this be "A voltage stabilizer was supplied with the freezer to prevent temperature fluctuation ...? I think so.

Lines 94-95: frying or oven-baking.

Also, why thaw the nuggets before cooking? This is generally not the recommended practice, as thawing affects the structural integrity and affects oil uptake during frying. The general practice is to cook nuggets in the frozen state without thawing

Lines 98-105: Some clarification is needed. Please clarify "one part of the fish nuggets" - need more detail for others to reproduce the work. Is this the 26 nuggets? Please give the quantity (number or weight). Also, why use 160±5 °C as the frying temperature? Most commercial temperatures range from 170 - 185 °C.

Also, considering the study used a 'fryer pan' - can this be termed 'deep frying'. There are specific fryers for deep-frying. What was the size (L) of the fryer, what was the oil volume (L), and the food to oil ratio? That will inform if the process is deep frying.

Line 108: Please state the method numbers for each of the methods.

Line 109: Do you mean "the equation adopted from Bonfim et al.'?

Line 112-113: Suggested revision “Lipid was extracted from processed fish nuggets ...... and used for the determination of ....... acetic acid-chloroform mixture (3:2 v/v) was …”

Lines 114-116: Is this much detail required for the PV test? This is a well-established method that is also present in the AOAC method book. I don't think you need this detail. Also, there is no need for vigorous shaking on adding the AA-Chloroform mixture to the oil/fat as it dissolves just fine. Vigorous shaking is required on adding the saturated KI to the mixture. Also, there is always an incubation period (typically 1- 2 minutes in the dark), then the addition of the deionized water to quench the reaction to titration.

Also, should it be the iodine released into the chloroform layer?

Lines 118-121: The method employed for the FFA analysis is not one of the currently used methods for oil tests as per AOAC, AOCS, ISO. Was there any reason to use this method instead of the currently known methods?

Was the method based on Lowry and Tinsley's method? If so, then first state that the FFA was quantified using the Lowry and Tinsley method [19].

Lines 124-125: Revision: "The color L*a*b color parameters of processed fish nugget samples were examined with a Hunterlab ...."

Line 126: Was it just a nugget or nuggets? Did each weight 25 g? - just using a nugget can introduce bias into the analysis.

Also - suggested revision - ".... placed in a pre-weighted centrifuge tube

Line 127: .... using an RC-5B ...

Line 128: Please clarify this or provide additional information - "The sample was removed and weighed (Wp)" - how was this done? E.g., - separation of pellet and precipitate by decantation?

Line 130: Equation 2 - is the "Wt" needed in the equation? I don't think so. 1) they cancel out when you expand the variables - hence, no need to add to 'Ws' and 'Wp’. 2) were the centrifuge tubes 'pre-weighed and tared? What does the 'weighed centrifuge tube mean? - Please consider Line 134: Texture analysis was performed using ....Line 144: Please provide details about the SPSS program (version, developer, city)

Line 151-152: It is of course expected to see higher fat levels for fried food.

Lines 152-154: Please correct this mechanism of oil uptake. Osmosis is the movement of a solvent/liquid across a selective semi-permissible membrane. The fish nuggets lack such membranes. The oil uptake is largely by absorption via porous pores and capillaries.

Lines 156-159: Revision '.... carbohydrate content decreased with associated ....... Moisture loss, due to dehydration, during frozen storage may explain the decrease in moisture content of the cooked nuggets."

Lines 159-161: This can be a problem from a formulation and quality viewpoint - ideally, the product should maintain such features throughout the shelf-life, hence a loss of breadcrumbs maybe affect the physical perception of the product.

Lines 161-162: Protein content started to change from Day 15 as per Table 1.

Line 163: It should be noted that this increase/decrease is only relative when comparing these two cooking methods - more uptake with frying, invariably relative protein content drops to account for mass balance

Line 165-166: The carbohydrate value on Day 60 for oven-baked seems like an outlier.

Line 172: ......fried and oven-baked fish nuggets (Table 1), respectively.

Line 174-175: Delete 'Protein, fat, 174 and carbohydrate contents contributed to the energy value." - This is already stated in the method where the energy value is calculated

Lines 176-177:  ......deep-fried nuggets are due to their...

Also, the protein and carbohydrate values are irrelevant here because the only distinguishing nutrient from the oven-baked is fat. Please revise to remove the remove reference to protein and carbohydrate because these are lower in the fried nugget compared to the oven-baked.

Lines 180-181: Table 1 - Maybe clarify the upper- and lower-case letters to state what they are comparing - between storage times and cooking method.

Lines 186-187: Thermal hydrolysis is incorrect. Thermal - heat, and hydro - water are two separate agents. A better term is 'thermolysis' of 'thermal degradation'. Alternatively, both thermal and hydrolytic degradation. For Day 0, thermal degradation might be a better term

Lines 188-189: Where is the data? Is there data on the samples before the cooking process or this is just inferential from the cooking data?

Line 188: Please explain 'FFA volatilization' - what does it mean? Of course, the dilution is reasonable mainly due to migration from the food's interior into the cooking oil provided new fresh oil is utilised for frying at those time points.

Did the authors measure the FFA of the unfried/unbaked nuggets at those time points before processing them?

Lines 190-191: Is the current protocol the best for the production of fish nuggets? Ideally, nuggets should have undergone some form of pre-treatment (partial cooking) before the frozen storage. Because the freezing process won't be adequate to inhibit enzymatic activity. Why were no pre-treatments included prior to frozen storage? Lipid degradation continues in frozen storage.

Lines 193-194: What evidence support the 'loss of volatile FFA' at the current cooking conditions? FFAs generally do not volatilize (of course under correct conditions of temperature and pressure - boiling points of fatty acids at standard conditions range from about 200 to 450 degrees Celsius) - the volatile carbonyl compounds do - aldehyde and ketones.

There is less evidence to support this - because it could be that the FFA had already reached a level in the frozen samples before frying and the additional cooking method can only contribute a certain amount and is not necessarily due to loss of 'volatile FFA'. This is slightly complicated because, to some extent, the frying oil will contribute to the fried nugget FFA. Please revisit this explanation - might be best to remove it.

Lines 197-199: The bars are not evident in the figure. Also, improve lines 198-199 for meaning

Line 203: Delete 'period'

Suggested revision "...Results revealed that the PV of cooked samples increased with storage time ..."

Lines 207-208: Is this the first study to observe this or there are other studies? Or will it be best to say, "the current study is consistent with previous studies"?

Line 209: ".... catfish fillets had lower PV compared to baked samples."

Line 210: ... Several studies postulate that ...

Line 211: .... decomposed into various secondary oxidation species/products. Please provide references for some of these studies.

Line 212: Considering, the heating temperatures are the same, and longer heating for oven-baking - one will expect a similar decomposition of the primary oxidation product? - Any explanation? Could this also be due to dilution due to mass transfer of the frying oil into the food as evidenced by Table 1 - the rice bran oil has better stability than the fish oil - a fatty acid analysis could have verified this.

Suggested revision ‘... Also, the antioxidants present in the rice bran oil ... and tocotrienols’

Line 216: ... of fried dough during storage.

Lines 217-218: Should this be 'stability against oxidative rancidity'? - Rancidity can be hydrolytic or oxidative

Suggested revision: '... A rancid flavor is perceptible at....’

Lines 221-222: Rancidity is a mechanism, not a product.

Revision: ".... shielded the fish nuggets from going rancid during the storage period"

Line 233: Section 3. The most distinguishing value in Table 2 is the 'a' values between fried and baked nuggets - but this has not been discussed

Lines 227-228: Are the differences for the 'L' and 'b' values the cooking methods that important as they are in the same ball-pack and not significant. However, the main effect is seen in the 'a' values.

Line 243-235: How so? the evidence is not strong to support this. Note that the rice bran oil has near deep yellow and absorption potentially contribute to the colour of the food - it could just be due to that and not so about greater Mallard/ caramelisation reaction with frying than baking - please support this with relevant literature.

Lines 236-267: There is no evidence to support and contrast this. The low protein content could mostly be due to the oil absorption and not so due to the Maillard reaction.

Line 238-239: Note that this is subject to consumer preference and depend on the food type

Lines 243-244: This is understandable for carbohydrates but not for protein. Was protein analysed in the uncooked products during storage? Also, protein increased as shown in Table 1.

Lines 252-254: How important is this - as one may argue this is expected because one was by wet processing (frying) and the oil by drying processing (baking) - the wet process will intuitively retain and express more fluid. Maybe the main effect is the storage time with expression increase with time. No explanation has been given for the increasing TEF with storage time

Line 257: Use 'is responsible' not 'may be'

Lines 258-259: How does this fit into the current discussion. It seems to be an isolated point.

Line 262-263: There is limited evidence in the current study to support this statement. Consider using the published literature to support this statement.

Lines 263-265: Not necessarily true. Account for the mass balance calculation - the frying led to greater moisture migration compared to the baking process. More so, the migration of moisture results in oil uptake to fill the spaces previously occupied by the water molecules (table 1).

Hence, the high oil/fat content in the fried products means, the protein will drop on a mass balance basis. And may not necessarily be due to the higher protein content in the baked product. Is water holding capacity the same as moisture content?  What accounts for the increasing TEF with storage time?

Lines 271-278: How does the relatively low TEF contribute to lower hardness?

Lines 272-273: What was these lower hardness values compared to?

Lines 274-275: Rephrase this statement - because these are known facts that are well reported in the literature by many researchers and not only Yong et al. - E.g., "Cooking foods causes starch gelatinization, protein denaturation and water evaporation."

Lines 275-276: The water content variances between the two processes are about 4-5%. Also, ideally, you will expect the free water to be expelled before bound water. What evidence suggests that bound water is expelled? Also, is a random aggregation of proteins absent in baking? And why? It should also be noted that the characteristics of the crust can also influence that hardness value

Lines 282-288: The degree of change were similar for springiness and cohesiveness at the different time points, but they have different interpretations here.

Lines 290-292: Table 3 - There was a significant variance between chewiness for fried ad baked nuggets that have not been discussed - it is the most difference in the table besides hardness

Line 317:  Suggested revision ‘... lower the initial microbial load’.

Lines 328-329: Better to state 'free fatty acids and peroxide values" because lipolysis and lipid oxidation are broader terms

line 327: ... influenced the overall quality ....

Line 330: In general, deep frying ....

Line 336: Change 'worthy' to 'acceptable'

Line 337: Change 'items' to 'products'

Author Response

Reviewer 2

The manuscript reports on the ‘comparative effect of frying and baking on the chemical, physical and microbiological characteristics of frozen fish nuggets. There are relatively similar published works exploring similar themes – cooking effects on fish nuggets.  However, the main variances here is the geographical setting and the relevance to the target, and species of fish used for the study which lends to value-addition to this fish product – contributing to postharvest strategies. Overall, the conclusions are consistent with the results and discussions. Authors should consider adding a contextual relevance of cassava croaker to the geographical location in the introduction.

Ans: It was changed to “Cassava croaker or cassava fish (Pseudotolithus senegalensis) is an economically important marine fish of tropical countries, particularly in West Africa [5]. It belongs to high protein lean fish category (21.2% protein and 0.9% fat) and is a source of minerals (calcium, potassium, sodium, phosphorus, magnesium, iron, zinc, and manganese) [6]. Thus, this fish has good nutritive values and can contribute positively to human nutrition.

Furthermore, aspects of the manuscript require further clarification and correction. For example, the idea that during frying, the mass transfer of oil into the fried food is by ‘osmosis’ – which is inaccurate as osmosis requires a selective semi-preamble membrane. Likewise, of the explanations were speculative without available evidence to support them.

Ans: The term ‘osmosis’ was replaced by ‘diffusion’. The explanation with the reference regarding the oil absorption during deep frying was also added.

During deep-frying, water was evaporated and frying oil was undergone diffusion to the fish flesh, leading to a higher fat content in deep-fried fish than in oven-baked fish [24, 25]. The oil uptake is largely by absorption via porous pores and capillaries. Parkash Kochhar and Gertz [26] proposed that the mechanism of oil absorption during deep frying is dependent on the amount of water removed and the way in which this moisture is lost. Once the fried product is taken from the hot oil and begins to cool, oil uptake is primarily a surface-related phenomenon caused by a competition between drainage and suction into the porous crust [27].

Specific comments are listed below.

Line 14: Revise to " The moisture, protein, fat, ash, and carbohydrate contents varied ...."

Ans: Done.

Line 15-16: This is expected considering the cooking method was frying with oil

Ans: Yes, however we must present our findings in an abstract. As a result, this sentence was stated.

Lines 27-28, This statement is not clear. Please revise. The first part is fine but the second part starting with food products with identified ...." is not clear and reads incomplete

Ans: It was changed to “Globally, there has been a gradual expansion of interest in health and healthy eating, which has prompted the food industry to produce or adapt food preparations in this direction [1].”

Lines 29-30: What does 'Unique chemical" properties mean? Fish has always been a protein source, hence, what makes that unique – Please include a reference for this statement - to support "promote consumer health and prevent some diseases"

Ans: It was changed to “Consumption of functional components found in marine fish-based products, notably n-3 polyunsaturated fatty acids like eicosapentaenoic acid and docosahexaenoic acid, is rapidly increasing in order to improve consumer health and prevent diseases like arte-riosclerotic and thrombotic disease [2].

Lines 32-33: What’s the meaning of 'creature protein'? Are they sources of supplements or are supplements? It should be the former - please review this statement

Ans: It was changed to “Additionally, fish contains a variety of nutrients, including protein, amino acids, vita-mins, and minerals [3, 4].

Lines 35-36: It is either a source or it isn't. Does it contain those minerals? If yes, then remove 'possible'

Ans: The word ‘possible’ was removed.

Lines 37-38: Consider this revision "Thus, this fish has good nutritive values and can contribute positively to human nutrition"

Ans: Done.

Line 50: Is 'food nutrition' a correct phrase?

Ans: It was changed to ‘nutritional value’

Line 53: ...are regarded as two of the oldest food processing methods

Ans: Done.

Lines 54-56: Please rephrase this to remove the references to 'field of sustenance' as the statement is well known in food processing, nutrition, and physiology. Basically, 'The type of cooking processing can damage important nutrients in foods.

Ans: It was changed to “In general, different types of cooking and processing can damage key nutrients in foods to varied degrees.

Line 60: Why technological properties? In food processing, technological properties refer to things such as emulsification, foaming properties etc.

The current study is largely about quality indices - physical, chemical and textural properties. Please revise

Ans: Thank you very much. It was changed to “Therefore, the present study aimed to investigate the effects of deep-frying and oven-baking on the quality characteristics of frozen (-18 °C/90 days) cassava fish nuggets.

Lines 66-67: Is it required to list all these analytical chemicals? Or best to state that "All analytical chemicals were purchased from Sigma-Aldrich (St. Louis, MO, USA)?

Ans: It was changed to “All analytical chemicals were purchased from Sigma-Aldrich (St. Louis, MO, USA).”

Lines 70-72: Consider this revision "Cassava croaker (P. senegalensis) weighing 1-2 kg were purchased from a local market in James Town,

Greater Accra, Ghana, West Africa." Remove 'along the coast ..."

Ans: Done.

Lines 72-73: What does 'off stacked' mean? Also, was it only one fish? Ideally, multiple fishes are a better sample representation.

... brought to the laboratories of ...

Please clarify this - If fish was purchased from a local supermarket, then clarify the statement in lines 72-73 - The fish was off stacked........

Should be 'a' local supermarket ...

Ans: Ten fish from a local market were used in this study. So, it was changed to “Ten cassava croaker (P. senegalensis) weighing 1-2 kg were purchased from a local market in James Town, Greater Accra, Ghana, West Africa. The fish were placed in ice with a fish/ice proportion of 1:2 (w/w) and brought to the Laboratory of Food Research Institute, Accra, Ghana within 50 min. Then, the fish were immediately washed with cold tap water (4 °C), gutted, clean, and filleted. The fish fillets were cut into 17-18 g square shapes (4×4×1.5 cm; width × length × height).

Line 75: ... gutted, cleaned, and filleted. The fish fillets were ......

Ans: Done.

Line 78: ...For each treatment, approximately 182 fillets ......

Please clarify this - You produced fillets that were cut into 17-18g square cubes/shapes - then they cease to be 'fillets'. Hence should the statement be "... approximately, 182 of cut fillet cubes were used.' There is no need to repeat the 17-18 g (already stated in line 75). Alternatively, give the cut fillets a name that you may refer to, e.g., product A.

Ans: It was changed to “For each treatment, approximately 182 of cut fillet cubes (CFC) were used.” The name 'CFC' referred to 'cut fillet cubes'.

Line 79: Delete 'approved'. How did these authors approve of such a method? An industry body or government agency has that responsibility.

Use '... using the method recommended by Duman and Kuzgun, and ...

Ans: This statement was removed due to Reviewer 1's suggestion that the nuggets be made with different ingredients.

Line 80: Note that 'you are using the cut cubes not the fillets - Fillets are thin cuts lengthwise along the bone, however, they have been cut into cubes, hence, no longer fillets

Ans: The term “CFC” was used.

Lines 81-82: All these ingredients are seasonings, hence clarify '27.8 f of complete seasoning' - what is it? On-the market ready-made composite seasoning ingredient? maybe include some details - brand/product name. Also, is it 15.5 g of lemon juice? or the whole lemon? Please indicate which part.

Ans: It was changed to “CFC were mixed with 70.6 g of garlic, 96.7 g of ginger, 27.8 g of ready-made seasoning powder (KnorrÒ, Bangkok, Thailand), 7.5 g of black pepper, 3.2 g of parsley, 37.5 g of mustard, 15.5 g of lemon juice and 16.1 g of salt.”

Line 82: Please review the appropriateness of the use of the term 'fillet' in different sections of the manuscript - I think a better classification/term will help clarify the process better - that is - you start with the harvested fish - make fillet - cut into small pieces (dicing) - then mixing and seasoning etc.

Ans: Thank you very much. The term 'fillet' was double-checked across several sections of the manuscript.

Lines 85-86: Suggestion, 'The proximate composition of the uncooked fish nugget was 55.3 ....'

Should the uncooked be 'raw nuggets'? As they are intuitively the same

Ans: It was changed to “According to AOAC [15], the proximate composition of the raw nuggets was 55.3±0.9% moisture, 34.0±0.1% protein, 5.9±0.1% fat, 2.5±0.1% ash, and 2.3±0.8% carbohydrate.

Line 85: Revise - 'The nuggets were kept on a tray for 30 min ..."

Ans: Done.

Line 88:  ... were either oven-baked or deep-fried ... Using 'add' suggests the use baked and fried the same product

Ans: Done.

Lines 92: Should this be "A voltage stabilizer was supplied with the freezer to prevent temperature fluctuation ...? I think so.

Ans: Done.

Lines 94-95: frying or oven-baking.

Ans: Done.

Also, why thaw the nuggets before cooking? This is generally not the recommended practice, as thawing affects the structural integrity and affects oil uptake during frying. The general practice is to cook nuggets in the frozen state without thawing

Ans: Thank you very much. We'll take notice to follow your advice. However, the samples were partially thawed in this case to avoid overcooking or the creation of a crust on the surface if frozen samples were used. It might also be linked to the realistic if frozen nuggets were moved from the supermarket to the kitchen of the customer. Products can also be partially thawed.

Lines 98-105: Some clarification is needed. Please clarify "one part of the fish nuggets" - need more detail for others to reproduce the work. Is this the 26 nuggets? Please give the quantity (number or weight). Also, why use 160±5 °C as the frying temperature? Most commercial temperatures range from 170 - 185 °C.

Also, considering the study used a 'fryer pan' - can this be termed 'deep frying'. There are specific fryers for deep-frying. What was the size (L) of the fryer, what was the oil volume (L), and the food to oil ratio? That will inform if the process is deep frying.

Ans: It was changed to “One part of the fish nuggets (26 nuggets) was deep-fried in 5-L King rice bran oil (γ-oryzanol, 8000 mg/kg) (Thai Edible Oil Co., Ltd., Bangkok, Thailand) at a temperature of 160±5 °C [16] for 6 min in a 6-L Shuangchi electrical fryer pan (Guangzhou Shuangchi Dining Equipment Co., Ltd., Guangzhou, China) with a food to oil ratio of 1:10 (w/v).

The frying temperature was set at 160±5 °C following the previous report of Teruel et al. (2015).

[Teruel, M.R.; Garrido, M.D.; Espinosa, M.C.; Linares, M.B. Effect of different format-solvent rosemary extracts (Rosmarinus officinalis) on frozen chicken nuggets quality. Food Chem. 2015, 172, 40-46.]

In addition, Choe and Min (2007) reported that “frying is a process of immersing food in hot oil with a contact among oil, air, and food at a high temperature of 150 °C to 190 °C.”

[Choe, E.; Min, D.B. Chemistry of deep‐fat frying oils. J. Food Sci. 2007, 72, R77-R86.]

Line 108: Please state the method numbers for each of the methods.

Ans: Done.

Line 109: Do you mean "the equation adopted from Bonfim et al.'?

Ans: It was changed to “The energy value was determined using the equation (1) adopted from Bonfim et al.

Line 112-113: Suggested revision “Lipid was extracted from processed fish nuggets ...... and used for the determination of ....... acetic acid-chloroform mixture (3:2 v/v) was …”

Ans: It was changed to “Lipid was extracted from processed fish nuggets by the method of Bligh and Dyer [19] and used for the determination of FFA and PV. The FFA was quantified using the Lowry and Tinsley method [20]. Briefly, isooctane (5 mL) and 5% cupric acetate-pyridine (1 mL) were added to a fat sample (0.1 g) and vortexed for 90 s. The absorbance of the top phase was measured at 715 nm using a UV-Vis spectrophotometer (Shimadzu, Kyoto, Japan). The FFA content was estimated using a standard curve of oleic acid [20]. For PV, AOCS Method Cd 8b-90 [21] was used and reported as milliequivalents of free iodine (meq)/kg fat.

Lines 114-116: Is this much detail required for the PV test? This is a well-established method that is also present in the AOAC method book. I don't think you need this detail. Also, there is no need for vigorous shaking on adding the AA-Chloroform mixture to the oil/fat as it dissolves just fine. Vigorous shaking is required on adding the saturated KI to the mixture. Also, there is always an incubation period (typically 1- 2 minutes in the dark), then the addition of the deionized water to quench the reaction to titration.

Also, should it be the iodine released into the chloroform layer?

Ans: The PV test's details have been removed. It was revised to “Lipid was extracted from processed fish nuggets by the method of Bligh and Dyer [19] and used for the determination of FFA and PV. The FFA was quantified using the Lowry and Tinsley method [20]. Briefly, isooctane (5 mL) and 5% cupric ace-tate-pyridine (1 mL) were added to a fat sample (0.1 g) and vortexed for 90 s. The ab-sorbance of the top phase was measured at 715 nm using a UV-Vis spectrophotometer (Shimadzu, Kyoto, Japan). The FFA content was estimated using a standard curve of oleic acid [20]. For PV, AOCS Method Cd 8b-90 [21] was used and reported as mil-liequivalents of free iodine (meq)/kg fat.

Lines 118-121: The method employed for the FFA analysis is not one of the currently used methods for oil tests as per AOAC, AOCS, ISO. Was there any reason to use this method instead of the currently known methods?

Was the method based on Lowry and Tinsley's method? If so, then first state that the FFA was quantified using the Lowry and Tinsley method [19].

Ans: The FFA was quantified using the Lowry and Tinsley method [20].

Lines 124-125: Revision: "The color L*a*b color parameters of processed fish nugget samples were examined with a Hunterlab ...."

Ans: Done.

Line 126: Was it just a nugget or nuggets? Did each weight 25 g? - just using a nugget can introduce bias into the analysis.

Also - suggested revision - ".... placed in a pre-weighted centrifuge tube

Ans: It was changed to “For the TEF, cooked nuggets (each weight 25 g) (Ws) was placed in a pre-weighed centrifuge tube….

Line 127: .... using an RC-5B ...

Ans: Done.

Line 128: Please clarify this or provide additional information - "The sample was removed and weighed (Wp)" - how was this done? E.g., - separation of pellet and precipitate by decantation?

Ans: The pelleted samples were removed and weighed.

Line 130: Equation 2 - is the "Wt" needed in the equation? I don't think so. 1) they cancel out when you expand the variables - hence, no need to add to 'Ws' and 'Wp’. 2) were the centrifuge tubes 'pre-weighed and tared? What does the 'weighed centrifuge tube mean? - Please consider

Ans: Because the centrifuge tubes were pre-weighted and tared, Wt was removed from the equation.

Line 134: Texture analysis was performed using ....Line 144: Please provide details about the SPSS program (version, developer, city)

Ans: Done.

Line 151-152: It is of course expected to see higher fat levels for fried food.

Ans: Yes, it is correct. However, we must present our findings.

Lines 152-154: Please correct this mechanism of oil uptake. Osmosis is the movement of a solvent/liquid across a selective semi-permissible membrane. The fish nuggets lack such membranes. The oil uptake is largely by absorption via porous pores and capillaries.

Ans: Thank you very much. The term ‘osmosis’ was replaced by ‘diffusion’. The explanation with the reference regarding the oil absorption during deep frying was also added.

The oil uptake is largely by absorption via porous pores and capillaries. Parkash Kochhar and Gertz [26] proposed that the mechanism of oil absorption during deep frying is dependent on the amount of water removed and the way in which this moisture is lost. Once the fried product is taken from the hot oil and begins to cool, oil uptake is primarily a surface-related phenomenon caused by a competition between drainage and suction into the porous crust [27].

Lines 156-159: Revision '.... carbohydrate content decreased with associated ....... Moisture loss, due to dehydration, during frozen storage may explain the decrease in moisture content of the cooked nuggets."

Ans: Done.

Lines 159-161: This can be a problem from a formulation and quality viewpoint - ideally, the product should maintain such features throughout the shelf-life, hence a loss of breadcrumbs maybe affect the physical perception of the product.

Ans: It can happen in commercial products that are realistic. During freezing and cooking, some breadcrumbs may be lost. So, it was changed to “At the same time, the loss of breadcrumbs from the coating during frozen storage and cooking may cause the decrease in carbohydrate content of the cooked nuggets. Loss of coating can occur at any stage during the preparation or frozen storage [28].

Lines 161-162: Protein content started to change from Day 15 as per Table 1.

Ans: Thank you very much. It was changed to “The protein content was higher in oven-baked fillets than in deep-fried fillets from Day 15 to Day 90 of storage.”

Line 163: It should be noted that this increase/decrease is only relative when comparing these two cooking methods - more uptake with frying, invariably relative protein content drops to account for mass balance

Ans: Thank you. This reference was deleted and the new statement was given at the end of the first paragraph.

Line 165-166: The carbohydrate value on Day 60 for oven-baked seems like an outlier.

Ans: When comparing frying and baking at the same storage duration, there was no significant different.

Line 172: ......fried and oven-baked fish nuggets (Table 1), respectively.

Ans: Done.

Line 174-175: Delete 'Protein, fat, 174 and carbohydrate contents contributed to the energy value." - This is already stated in the method where the energy value is calculated

Ans: Done.

Lines 176-177:  ......deep-fried nuggets are due to their...

Ans: Done.

Also, the protein and carbohydrate values are irrelevant here because the only distinguishing nutrient from the oven-baked is fat. Please revise to remove the remove reference to protein and carbohydrate because these are lower in the fried nugget compared to the oven-baked.

Ans: Done.

Lines 180-181: Table 1 - Maybe clarify the upper- and lower-case letters to state what they are comparing - between storage times and cooking method.

Ans: It was changed to “Values are mean ± standard deviation from triplicate determinations. Different upper case letters in the same row (between storage times) and different lower case letters among samples (between cooking methods) at the same time indicate significant different (p < 0.05).

Lines 186-187: Thermal hydrolysis is incorrect. Thermal - heat, and hydro - water are two separate agents. A better term is 'thermolysis' of 'thermal degradation'. Alternatively, both thermal and hydrolytic degradation. For Day 0, thermal degradation might be a better term

Ans: It was changed to “thermal degradation”.

Lines 188-189: Where is the data? Is there data on the samples before the cooking process or this is just inferential from the cooking data?

Ans: It was from the cooking data. So, it was changed to “Results revealed that there was a slight increase in FFA throughout the storage period in both cooked samples (Fig. 1a).

Line 188: Please explain 'FFA volatilization' - what does it mean? Of course, the dilution is reasonable mainly due to migration from the food's interior into the cooking oil provided new fresh oil is utilised for frying at those time points.

Ans: The term “FFA volatilization” was deleted and it was changed to “Also, lower FFA values in deep-fried samples could be owing to dilution caused by migration from the food's interior into the cooking oil.

Did the authors measure the FFA of the unfried/unbaked nuggets at those time points before processing them?

Ans: Unfortunately, the FFA of the unfried/unbaked nuggets was not determined. Cooked nuggets were used to determine the FFA.

Lines 190-191: Is the current protocol the best for the production of fish nuggets? Ideally, nuggets should have undergone some form of pre-treatment (partial cooking) before the frozen storage. Because the freezing process won't be adequate to inhibit enzymatic activity. Why were no pre-treatments included prior to frozen storage? Lipid degradation continues in frozen storage.

Ans: Thank you so much for your great advice. Pre-cooking was not used for the fish nuggets in this investigation. In the future, we will use the pre-treatment protocol.

However, in the Introduction, we stated that “Commercially, the fish nuggets, either raw or pre-cooked, are generally in the frozen form [9, 10].

Lines 193-194: What evidence support the 'loss of volatile FFA' at the current cooking conditions? FFAs generally do not volatilize (of course under correct conditions of temperature and pressure - boiling points of fatty acids at standard conditions range from about 200 to 450 degrees Celsius) - the volatile carbonyl compounds do - aldehyde and ketones.

There is less evidence to support this - because it could be that the FFA had already reached a level in the frozen samples before frying and the additional cooking method can only contribute a certain amount and is not necessarily due to loss of 'volatile FFA'. This is slightly complicated because, to some extent, the frying oil will contribute to the fried nugget FFA. Please revisit this explanation - might be best to remove it.

Ans: Thank you very much. It was removed.

Lines 197-199: The bars are not evident in the figure. Also, improve lines 198-199 for meaning

Ans: It was changed to “Different upper case letters in the same cooking method (between storage times) and different lower case letters between cooking methods at the same time indicate significant different (p < 0.05).

Line 203: Delete 'period'

Ans: Done.

Suggested revision "...Results revealed that the PV of cooked samples increased with storage time ..."

Ans: Done.

Lines 207-208: Is this the first study to observe this or there are other studies? Or will it be best to say, "the current study is consistent with previous studies"?

Ans: It was changed to “The current study is consistent with previous studies that deep-fried nuggets had relatively lower PV than oven-baked nuggets.

Line 209: ".... catfish fillets had lower PV compared to baked samples."

Ans: Done.

Line 210: ... Several studies postulate that ...

Ans: Done.

Line 211: .... decomposed into various secondary oxidation species/products. Please provide references for some of these studies.

Ans: Done.

Line 212: Considering, the heating temperatures are the same, and longer heating for oven-baking - one will expect a similar decomposition of the primary oxidation product? - Any explanation? Could this also be due to dilution due to mass transfer of the frying oil into the food as evidenced by Table 1 - the rice bran oil has better stability than the fish oil - a fatty acid analysis could have verified this.

Ans: The statement was added. “Furthermore, as shown in Table 1, dilution owing to mass transfer of the frying oil into the food, as well as the fact that rice bran oil is more stable than fish oil [36], could all contribute to the lower PV of the deep-fried nuggets.”

Suggested revision ‘... Also, the antioxidants present in the rice bran oil ... and tocotrienols’

Ans: Done.

Line 216: ... of fried dough during storage.

Ans: Done.

Lines 217-218: Should this be 'stability against oxidative rancidity'? - Rancidity can be hydrolytic or oxidative

Ans: It was changed to “PV is normally used to survey stability against oxidative rancidity [34]. A rancid flavor is perceptible at a PV of 10-20 meq/kg fat [37].

Suggested revision: '... A rancid flavor is perceptible at....’

Ans: Done.

Lines 221-222: Rancidity is a mechanism, not a product.

Revision: ".... shielded the fish nuggets from going rancid during the storage period"

Ans: Done.

Line 233: Section 3. The most distinguishing value in Table 2 is the 'a' values between fried and baked nuggets - but this has not been discussed

Ans: The a* values were discussed as suggested.

Lines 227-228: Are the differences for the 'L' and 'b' values the cooking methods that important as they are in the same ball-pack and not significant. However, the main effect is seen in the 'a' values.

Ans: It was changed to “Throughout the storage period, the a* value of deep-fried nuggets was substantially higher than that of oven-baked nuggets (p < 0.05). The b* values of deep-fried nuggets were likewise greater than those of oven-baked nuggets. As a result, the deep-fried nuggets were a vivid orange-yellow color, but the oven-baked nuggets were a pale yellow tone.

Line 243-235: How so? the evidence is not strong to support this. Note that the rice bran oil has near deep yellow and absorption potentially contribute to the colour of the food - it could just be due to that and not so about greater Mallard/ caramelisation reaction with frying than baking - please support this with relevant literature.

Ans: The statement was added. “The lipid oxidation and Maillard reaction, according to Zhang et al. [29], are the most common and well-studied alterations that come into contact with changes in both frying oil and food material components. Maillard reactions, also known as non-enzymatic browning reactions, are a series of essential chemical changes that con-tribute to the color and odor of food during processing [40]. On the one hand, the presence of carbonyl group in the frying system is supplied by the creation of aldehydic compounds as a result of thermal oxidation. Frying materials, on the other hand, provided the amino group.

Also, the contribution of rice bran oil absorption on color of the food was given. “Also, the rice bran oil is a near-dark yellow color, and absorption could affect the color of the fried nuggets.

Lines 236-267: There is no evidence to support and contrast this. The low protein content could mostly be due to the oil absorption and not so due to the Maillard reaction.

Ans: The statement about protein was deleted.

Line 238-239: Note that this is subject to consumer preference and depend on the food type

Ans: It was changed to “According to Bechtel et al. [41], the most ideal color for fried products was a light brown color, depending on the food type.

Lines 243-244: This is understandable for carbohydrates but not for protein. Was protein analysed in the uncooked products during storage? Also, protein increased as shown in Table 1.

Ans: It was changed to “This may be due to the decreases in the substrates for non-enzymatic browning reaction, particularly carbohydrate, in both samples with increasing frozen storage time (Table 1).

Lines 252-254: How important is this - as one may argue this is expected because one was by wet processing (frying) and the oil by drying processing (baking) - the wet process will intuitively retain and express more fluid. Maybe the main effect is the storage time with expression increase with time. No explanation has been given for the increasing TEF with storage time

Ans: It was changed to “It can be observed that significant differences were observed among the two cooking methods during storage (p < 0.05). In addition, the TEF of both samples increased as storage time increased (p < 0.05). Therefore, the cooking methods and frozen storage periods affected the fluid retention capacity of the fish nuggets. The baked fish nuggets had relatively lower TEF than the deep-fried nuggets at all-time points (p < 0.05), which is expected because the wet process (frying) will intuitively retain and express more fluid than the dry process (baking).

Line 257: Use 'is responsible' not 'may be'

Ans: Done.

Lines 258-259: How does this fit into the current discussion. It seems to be an isolated point.

Ans: A statement was deleted.

Line 262-263: There is limited evidence in the current study to support this statement. Consider using the published literature to support this statement.

Ans: This statement was deleted.

Lines 263-265: Not necessarily true. Account for the mass balance calculation - the frying led to greater moisture migration compared to the baking process. More so, the migration of moisture results in oil uptake to fill the spaces previously occupied by the water molecules (table 1).

Hence, the high oil/fat content in the fried products means, the protein will drop on a mass balance basis. And may not necessarily be due to the higher protein content in the baked product. Is water holding capacity the same as moisture content?  What accounts for the increasing TEF with storage time?

Ans: Thank you very much. The new statement was given. “In comparison to baking, frying caused more moisture migration. More importantly, moisture migration causes oil uptake to fill the spaces previously occupied by water molecules (Table 1). Because the adsorbed oil is not strongly connected to the nugget structures, more TEF can be detected at all-time points.

Lines 271-278: How does the relatively low TEF contribute to lower hardness?

Ans: We do agree with the reviewer. So, the statement was changed to “The hardness of deep-fried nuggets was higher than baked nuggets till the end of the storage period (p < 0.05). According to Bechtel et al. [41], baked breaded and battered catfish had a lower hardness value than fried breaded and battered catfish. The crust formation may be linked to the greater hardness value in deep-fried nuggets.…….

Lines 272-273: What was these lower hardness values compared to?

Ans: It was changed to “According to Bechtel et al. [41], baked breaded and battered catfish had a lower hard-ness value than fried breaded and battered catfish.

Lines 274-275: Rephrase this statement - because these are known facts that are well reported in the literature by many researchers and not only Yong et al. - E.g., "Cooking foods causes starch gelatinization, protein denaturation and water evaporation."

Ans: Done.

Lines 275-276: The water content variances between the two processes are about 4-5%. Also, ideally, you will expect the free water to be expelled before bound water. What evidence suggests that bound water is expelled? Also, is a random aggregation of proteins absent in baking? And why? It should also be noted that the characteristics of the crust can also influence that hardness value

Ans: This statement was deleted and the crust formation was discussed.

Lines 282-288: The degree of change were similar for springiness and cohesiveness at the different time points, but they have different interpretations here.

Ans: It was changed to “Springiness and cohesiveness of both samples showed similar degrees of change at different time points. Springiness represents the deformation when the compressive force is taken away [49]. Cohesiveness represents the resistance of the internal bonds [48]. Springiness of the deep-fried nuggets was higher than oven-baked nuggets (p < 0.05) whereas the cohesiveness values of both samples were not different (p > 0.05). Based on the findings, oven-baked fish nuggets were softer compared to the deep-fried nuggets.………….”

Lines 290-292: Table 3 - There was a significant variance between chewiness for fried ad baked nuggets that have not been discussed - it is the most difference in the table besides hardness

Ans: The discussion about the chewiness was given.

Line 317:  Suggested revision ‘... lower the initial microbial load’.

Ans: Done.

Lines 328-329: Better to state 'free fatty acids and peroxide values" because lipolysis and lipid oxidation are broader terms

Ans: The terms “FFA” and “PV” were used.

line 327: ... influenced the overall quality ....

Ans: Done.

Line 330: In general, deep frying ....

Ans: Done.

Line 336: Change 'worthy' to 'acceptable'

Ans: Done.

Line 337: Change 'items' to 'products'

Ans: Done.

Round 2

Reviewer 1 Report

 The authors adequately revised the manuscript following my suggestions

The English language should be improved.

Author Response

Reviewer 1

The authors adequately revised the manuscript following my suggestions

The English language should be improved.

Ans: Thank you very much. The English language was rechecked carefully using Quillbot, a paraphrasing tool.

Reviewer 2 Report

Below are some minor issues for the authors.

Lines 34-35: ...disease including arteriosclerosis and thrombosis......

Lines 37-38: Revision 'Additionally, fish is a rich source of proteins, amino acids, vitamins and minerals.'

Line 63: varying

Line 83: ...The fishes ...

Line 95: Is this a mustard paste or oil or seed?

Lines 107-109: This is a significant variance from the draft 1. How do you thaw at 4 °C to a core temperature of 7 to -5 °C? Did you measure the core temperatures? All was that necessary. Or maybe state that the ......partially thawed for 60 min at 4 °C. Delete the reference to a core temperature.

Lines 126-128: Is the 'number' needed? AOAC method 950.46 etc.

Line 153: How many nuggets were used? Example - Two suggest (25 g each) (Ws) were placed in

Lines 178-180: ... The moisture ....... contents varied ....

Line 183: Suggestion " ... water was evaporated and partially replaced by the diffusion of the frying oil into the fish nugget"

Lines 185-190: I am not sure you need this. Since the current analysis is not about the detailed mechanism. All you need is in line 124. You may delete this new insert.

Lines 209-210: ... thermal degradation of lipids. During cooking processes, a series of complex reactions such as ......

Lines 217-218: This statement is like lines 218-221. Delete lines 217-218 and retain the other - they serve the same meaning.

Lines 219-220: .... higher fat content. Author: Randy Subject: Highlight Date: 16/12/2021 1:38:57 PM Line 228: ...cooking methods during storage (Fig 1a).

Line 237: Revision ' ...degradation of lipids during frying and baking of fish nuggets. ... ' Line 257: Delete 'also' Line 258: ... had lower PV compared ...

Lines 259-260: Delete 'can be quickly' because this mechanism is well documented

Lines 261-262: Delete this new insert. Rather as part of line 260 state examples of those secondary oxidation products - These volatile and non-volatile carbonyl compounds. That will be more relevant to the discussion than the current lines 261-262. Example: ... into various secondary oxidation products such as aldehyde, ketones etc... (check the literature to update). or some specific ones regarding lipids in fish and rice bran oil

Line 263: ......low PV compared to oven-baking.

Line 266: ... contained approximately 8,000 mg/kg

Line 277: Delete period

Line 288: Change 'substantially' to 'significantly'

Lines 294-297: Suggestion 'According to Zhang et al. [29], lipid oxidation and Maillard reaction are two of the most common and well-studied changes in both the frying oil and fried food.'

Lines 297-301: Delete this new insert as it does not add relevance to the current analysis.

Lines 301-303: Hence, it may be postulated that, frying generated more ........

Line 305: .... oil has a dark yellow color ....

Lines 327-328: Delete 'It can be observed that. Start with 'Significant differences ........ among the two cooking methods (p < 0.05).

Line 329-330: Delete 'Therefore .... of the fish nuggets.’ It only repeats previous statements.

Line 329: ... samples increased with storage time ...

Lines 344-348: ... more moisture migration from the cooked nuggets. The loss of moisture from the fried nuggets causes oil .... Is 'adsorbed' or 'absorbed' is the correct terminology? Note that 'adsorbed' is only a surface phenomenon - please consider this clarification

Lines 348-349: Change 'tight and immobilized' to 'bound'.

Lines 353-354: Delete 'Heat and mass transfer ....... of these phenomena' as it adds little to the context

Line 355: .... between the molars ....

Line 356: .... than the baked nuggets throughout the storage period (p < 0.05). .....

Lines 366-370: Revise to 'Regardless of the cooking process used in this study, moisture content was negatively correlated with hardness, whereas protein and fat levels were positively correlated with hardness (R2 = 0.6203-0.7786).'

Lines 391-392: It was previously stated that 'hardness' was linked to crust formation, line 360. If chewiness is also linked to crust formation, what is the relationship between chewiness and hardness? This should be articulated clearly because crust formation alone won't define chewiness – as you stated in lines 388-389, chewiness is a derived textural parameter and depends on the three previous parameters.

Lines 392-394: The stated reference [50] is about "Reactive intermediates and carbohydrate fragmentation in Maillard chemistry" and does not discuss 'chewiness'. Please insert a more appropriate reference.

Line 395: .... such as deep-fried ....

Line 402: .... subsequent cooking of fish nuggets.

Line 437: Delete 'period'

References: Sixty-four (64) references are very substantial for such a research paper/article.

Author Response

Reviewer#2

Below are some minor issues for the authors.

Lines 34-35: ...disease including arteriosclerosis and thrombosis......

Ans: Done.

Lines 37-38: Revision 'Additionally, fish is a rich source of proteins, amino acids, vitamins and minerals.'

Ans: Done.

Line 63: varying

Ans: Done.

Line 83: ...The fishes ...

Ans: Done.

Line 95: Is this a mustard paste or oil or seed?

Ans: It is a mustard paste.

Lines 107-109: This is a significant variance from the draft 1. How do you thaw at 4 °C to a core temperature of 7 to -5 °C? Did you measure the core temperatures? All was that necessary. Or maybe state that the ......partially thawed for 60 min at 4 °C. Delete the reference to a core temperature.

Ans: Thank you very much. It was changed to “Frozen samples were taken for deep-frying or oven-baking, after partially thawed for 60 min at 4 °C, at 15 days intervals and further analyses were carried out.”

Lines 126-128: Is the 'number' needed? AOAC method 950.46 etc.

Ans: The numbers were removed.

Line 153: How many nuggets were used? Example - Two suggest (25 g each) (Ws) were placed in

Ans: It was changed to “For the TEF, two cooked nuggets (25 g each) (Ws) were placed in a pre-weighed centrifuge tube and then centrifuged (2,960×g/25 °C/3 min) using an RC-5B plus centrifuge (Sorvall, Norwalk, CT, USA).

Lines 178-180: ... The moisture ....... contents varied ....

Ans: Done.

Line 183: Suggestion " ... water was evaporated and partially replaced by the diffusion of the frying oil into the fish nugget"

Ans: Done.

Lines 185-190: I am not sure you need this. Since the current analysis is not about the detailed mechanism. All you need is in line 124. You may delete this new insert.

Ans: It was deleted.

Lines 209-210: ... thermal degradation of lipids. During cooking processes, a series of complex reactions such as ......

Ans: Done.

Lines 217-218: This statement is like lines 218-221. Delete lines 217-218 and retain the other - they serve the same meaning.

Ans: Done.

Lines 219-220: .... higher fat content. Author: Randy Subject: Highlight Date: 16/12/2021 1:38:57 PM Line 228: ...cooking methods during storage (Fig 1a).

Ans: Done.

Line 237: Revision ' ...degradation of lipids during frying and baking of fish nuggets. ... ' Line 257: Delete 'also' Line 258: ... had lower PV compared ...

Ans: Done.

Lines 259-260: Delete 'can be quickly' because this mechanism is well documented

Ans: Done.

Lines 261-262: Delete this new insert. Rather as part of line 260 state examples of those secondary oxidation products - These volatile and non-volatile carbonyl compounds. That will be more relevant to the discussion than the current lines 261-262. Example: ... into various secondary oxidation products such as aldehyde, ketones etc... (check the literature to update). or some specific ones regarding lipids in fish and rice bran oil

Ans: Done.

Line 263: ......low PV compared to oven-baking.

Ans: Done.

Line 266: ... contained approximately 8,000 mg/kg

Ans: Done.

Line 277: Delete period

Ans: Done.

Line 288: Change 'substantially' to 'significantly'

Ans: Done.

Lines 294-297: Suggestion 'According to Zhang et al. [29], lipid oxidation and Maillard reaction are two of the most common and well-studied changes in both the frying oil and fried food.'

Ans: Done.

Lines 297-301: Delete this new insert as it does not add relevance to the current analysis.

Ans: Done.

Lines 301-303: Hence, it may be postulated that, frying generated more ........

Ans: Done.

Line 305: .... oil has a dark yellow color ....

Ans: Done.

Lines 327-328: Delete 'It can be observed that. Start with 'Significant differences ........ among the two cooking methods (p < 0.05).

Ans: Done.

Line 329-330: Delete 'Therefore .... of the fish nuggets.’ It only repeats previous statements.

Ans: Done.

Line 329: ... samples increased with storage time ...

Ans: Done.

Lines 344-348: ... more moisture migration from the cooked nuggets. The loss of moisture from the fried nuggets causes oil .... Is 'adsorbed' or 'absorbed' is the correct terminology? Note that 'adsorbed' is only a surface phenomenon - please consider this clarification

Ans: Done. It is ‘absorbed’.

Lines 348-349: Change 'tight and immobilized' to 'bound'.

Ans: Done.

Lines 353-354: Delete 'Heat and mass transfer ....... of these phenomena' as it adds little to the context

Ans: Done.

Line 355: .... between the molars ....

Ans: Done.

Line 356: .... than the baked nuggets throughout the storage period (p < 0.05). .....

Ans: Done.

Lines 366-370: Revise to 'Regardless of the cooking process used in this study, moisture content was negatively correlated with hardness, whereas protein and fat levels were positively correlated with hardness (R2 = 0.6203-0.7786).'

Ans: Done.

Lines 391-392: It was previously stated that 'hardness' was linked to crust formation, line 360. If chewiness is also linked to crust formation, what is the relationship between chewiness and hardness? This should be articulated clearly because crust formation alone won't define chewiness – as you stated in lines 388-389, chewiness is a derived textural parameter and depends on the three previous parameters.

Ans: Thank you very much. It was changed to “It has been reported that the existence of crust impacts the food's mechanical qualities, as well as its texture and acceptability [50]. As a result, the chewiness and hardness of deep-fried nuggets may be related to crust development. The chewiness of both nuggets changed in a similar way with the hardness, as it did in other products such as deep-fried battered and breaded fish balls [51] and gluten-free breads [52].

Lines 392-394: The stated reference [50] is about "Reactive intermediates and carbohydrate fragmentation in Maillard chemistry" and does not discuss 'chewiness'. Please insert a more appropriate reference.

Ans: It is a technical error for the running of reference from the previous round. Actually, reference 50 “Bordin, K.; Tomihe Kunitake, M.; Kazue Aracava, K.; Silvia Favaro Trindade, C. Changes in food caused by deep fat frying-A review. Archivos Latinoamericanos de Nutricion. 2013. 63, 5-13.” also discussed about “chewiness”.

Line 395: .... such as deep-fried ....

Ans: Done.

Line 402: .... subsequent cooking of fish nuggets.

Ans: Done.

Line 437: Delete 'period'

Ans: Done.

References: Sixty-four (64) references are very substantial for such a research paper/article.

Ans: We did our best to get references to back up our statements. There are presently a total of 58 references. In addition, there was a request in the first round to include more references.
